



**Effects of Cloud Condensation Nuclei and Ice Nucleating Particles on Precipitation**
**Processes and Supercooled Liquid in Mixed-phase Orographic Clouds**
Jiwen Fan[1*], L. Ruby Leung[1], Daniel Rosenfeld[2], Paul J. DeMott[3]
[1] Atmospheric Science & Global Change Division, Pacific Northwest National Laboratory,
Richland, WA 99352
[2] Institute of Earth Sciences, The Hebrew University of Jerusalem, Jerusalem, 91904 Israel
[3] Department of Atmospheric Science, Colorado State University, Fort Collins, Co, 80523
• Corresponding author:
jiwen.fan@pnnl.gov



**Abstract**
22         How orographic mixed-phase clouds respond to the change of cloud condensation nuclei

(CCN) and ice nucleating particles (INPs) are highly uncertain. The main snow production
mechanism in warm and cold mixed-phase orographic clouds (referred to as WMOC and CMOC,
respectively, distinguished here as those having cloud tops warmer and colder than -20°C) could
be very different. We quantify the CCN and INP impacts on supercooled water content, cloud
phases and precipitation for a WMOC and a CMOC case with a set of sensitivity tests. It is found
that deposition plays a more important role than riming for forming snow in the CMOC, while
the role of riming is dominant in the WMOC case. As expected, adding CCN suppresses
precipitation especially in WMOC and low INP. However, this reverses strongly for CCN >
1000 cm$^{-3}$. We find a new mechanism through which CCN can invigorate mixed-phase clouds
over the Sierra Nevada Mountains and drastically intensify snow precipitation when CCN
concentrations are high (1000 cm$^{-3}$ or higher). In this situation, more widespread shallow clouds
with greater amount of cloud water form in the valley and foothills, which changes the local
circulation through more latent heat release that transports more moisture to the windward slope,
leading to much more invigorated mixed-phase clouds over the mountains that produce higher
amounts of snow precipitation. Increasing INPs leads to decreased riming and mixed-phase
fraction in the CMOC but has the opposite effects in the WMOC, as a result of liquid-limited and
ice-limited conditions, respectively. However, it increases precipitation in both cases due to an
increase of deposition for the CMOC but enhanced riming and deposition in the WMOC.
Increasing INPs dramatically reduces supercooled water content and increases the cloud
glaciation temperature, while increasing CCN has the opposite effects with much smaller
significance.



## 1. Introduction

Snowpack in the Sierra Nevada Mountains is California's largest source of fresh water.
Understanding the factors contributing to snow precipitation over the mountains has important
implications to predicting the hydrology and local climate of the western U.S. This has motivated
a series of CalWater field campaigns carried out since 2009 to improve understanding of
processes influencing precipitation and water supply in California. Supercooled liquid occurred
commonly in clouds over the Sierra Nevada during the cold season (Rosenfeld et al., 2013).
Closely linked to precipitation is the distribution of cloud liquid and ice phases, which may be
influenced by supercooled liquid commonly occurring in orographic clouds (Rosenfeld et al.,
2013). Besides precipitation, cloud radiative forcing and cloud feedback in the climate system
are also highly dependent on cloud phases because of the very different radiative effect of liquid
and ice particles. Hence understanding the key processes and factors impacting cloud phases is
critical, but our lack of understanding and ability to model supercooled liquid and cloud phases is
limiting skillful predictions at weather and climate time scales.
Many factors such as large-scale dynamics, solar heating, and aerosol particles can
impact cloud properties and precipitation over the Sierra Nevada Mountains (Shen et al., 2010;
Rosenfeld et al., 2008). Atmospheric rivers (ARs) are one of the primary large-scale dynamical
features that bring large amount of water vapor from tropics to the U.S. west coast, and can
create extreme rainfall and floods (Bao et al 2006; Ralph et al. 2011; Neiman et al. 2010).
Aerosols in the atmosphere can modify cloud microphysical processes and potentially alter the
location, intensity, and type of precipitation (Tao et al., 2012) by acting as cloud condensation
nuclei (CCN) or ice nucleating particles (INPs). In California, pollution aerosols from the
densely populated coastal plains and the Central Valley may be incorporated into the frontal
airmass before orographic ascent and influence precipitation in the Sierra Nevada Mountains




(Rosenfeld and Givati, 2006). Long-range transported aerosols (mainly dust particles) have also
been found to have a potential influence on clouds and precipitation in the winter and spring
seasons (Uno et al., 2009; Ault et al., 2011; Creamean et al., 2013).
Aerosol impacts can strongly depend on aerosol properties, but also dynamics, and
thermodynamics. Many studies have shown that CCN can reduce warm rain precipitation from
orographic clouds by reducing the efficiency of cloud droplets conversion into raindrops (e.g.,
Lynn et al., 2007; Rosenfeld and Givati, 2006; Jirak and Cotton, 2006) and can reduce snowfall
precipitation due to reduced riming efficiency (Lowenthal et al., 2011; Rosenfeld et al., 2008).
However, some recent studies show a possibility of increased precipitation by CCN in
orographic mixed-phase clouds (Fan et al., 2014; Xiao et al., 2015). Other studies have shown
that CCN may not have significant effect on the total precipitation, but rather shift precipitation
from the windward to leeward slope; a so-called "spillover effect" (Saleeby et al., 2011; 2013).
By acting as INPs, aerosols can enhance ice growth processes such as deposition and riming and
thereby significantly increase snow precipitation (Fan et al., 2014). Both observational and
modeling studies have shown that long-range transported dust particles can enhance orographic
precipitation in California by serving as INPs (Ault et al., 2011; Creamean et al., 2013; Fan et al.,

2014).

Besides precipitation, aerosols may have significant impacts on cloud phase and
supercooled water content (SCW) in the mixed-phase clouds, which directly change cloud
radiative forcing and Earth's energy balance. Modeling studies have shown that CCN tend to
increase SCW via the processes such as suppressed warm rain and/or reduced riming efficiency
(Khain et al., 2009; Ilotoviz et al., 2016; Saleeby et al., 2013). A recent observational study
corroborated that increasing CCN decreases the cloud glaciation temperature and thus increases





the abundance of the mixed-phase regime (Zipori et al., 2015). With abundant INPs such as dust
particles, cloud glaciates at a much warmer temperature (Rosenfeld et al., 2011; Zipori et al.,
2015). It is found that commonly occurring supercooled water in the clouds near the coastal
regions of the western U.S. is associated with low CCN and limited INP conditions (Rosenfeld et
al., 2013).

Recent evaluation of the Community Atmosphere Model version 5 (CAM5) with the

Cloud-Aerosol Lidar and Infrared Pathfinder Satellite Observation (CALIPSO) satellite data
showed that the model has insufficient liquid cloud and excessive ice cloud from the mid-
latitudes to the polar regions, and liquid deficit bias maximizes over the Southern Ocean where
supercooled water is prevalent (Kay et al., 2016).  For cloud model simulations with cloud-
resolving models, ice nucleation parameterizations often need to be modified in order to produce
the mixed-phase clouds in the Arctic region (Fan et al., 2009; Fridlind et al., 2007). Considering
many microphysical processes are sensitive to aerosol types (CCN or INP), temperature, and/or
supersaturation (e.g., deposition growth), aerosol impacts on cloud phase can be complicated,
depending on cloud dynamics and thermodynamics. Our current understanding of cloud
microphysical processes impacting SCW and cloud phase in different meteorological
environments is poor. Therefore, it is important to conduct process-level studies to improve our
understanding.

Fan et al. (2014) conducted a study for two mixed-phase orographic cloud cases with

different cloud temperatures and showed different significance of the CCN and INP impacts
between the two cases. The two cases are February 15-16, 2011 (FEB16), and Mar 1-2, 2011
(MAR02). FEB16 has a cloud top temperature as cold as -32°C while the cloud top temperature
of MAR02 is generally warmer than -20°C. The temperature differences at the same altitude



between the two cases are about 6-10°C. For these reasons, we will herein refer to them as cold
mixed-phase orographic clouds (CMOC) and warm mixed-phase orographic clouds (WMOC),
respectively. The main snow-forming mechanism in warm and cold mixed-phase orographic
clouds could be very different and lead to different precipitation response to changes of CCN and
INP. Following Fan et al. (2014) this study aims to (1) understand the dominant ice growth
processes in these two mixed-phase cloud systems; (2) quantify the response of precipitation to
the changes of CCN and INP over a wide range from extremely low to extremely high
concentrations, and (3) examine CCN and INP impacts on SCW and cloud phases. The same
WRF model with the spectral-bin microphysics (SBM) as used in Fan et al. (2014) is employed.
Ice nucleation is parameterized in dependence on mineral dust/biological particle concentrations
on the basis of observational evidence. To better realize our science goals, the simulation
resolution is further increased to be 1-km and the simulations are driven with the 2-km resolution
baseline simulation from Fan et al. (2014).

**2. Model Description and Simulation Design**
**2.1 Model description**

As in Fan et al. (2014), simulations are performed using WRF version 3.1.1 developed at

the National Center for Atmospheric Research (NCAR) (Skamarock et al., 2008) coupled with a
spectral-bin microphysics (SBM) model (Khain et al., 2009; Fan et al., 2012). The SBM is a fast
version of the full SBM described by Khain et al. (2004), in which ice crystal and snow
(aggregates) in the full SBM are calculated based on one size distribution with separation at 150
μm. ice crystal and snow are referred to as low-density ice. Graupel and hail in the full SBM are





grouped as high-density ice, represented with one size distribution without separation. More
details about SBM that we used in this study can be found in Fan et al. (2014).
As discussed in Fan et al. (2014), hereafter referred to as FAN2014, the ice nucleation
parameterizations in the SBM used for this study have been modified. A new ice nucleation
parameterization of DeMott et al. (2015; cited as DeMott et al., 2013 in FAN2014 before the
parameterization was published) was incorporated to SBM to investigate the impacts of dust as
INPs. The parameterization connects nucleated ice particle concentration under a certain
atmospheric condition with aerosol particle number concentration with diameter larger than 0.5
µm ($n_{a>0.5\mu m}$ in Eq. 2 of DeMott et al., 2015), which is referred to as INP concentration. In
FAN2014, the aerosol particles that are connected with the DeMott et al. (2015) parameterization
are referred to as "dust/bio" (from single particle mass spectral composition measurements), and
are based on observations from the Passive Cavity Aerosol Spectrometer Probe (PCASP) for
particles with diameter larger than 0.5 µm from clear-sky aircraft data. Note that the actual INP
number concentration in the DeMott et al. (2015) parameterization includes an exponential
temperature dependence that acts on aerosol concentration, and that the exponent on aerosol
concentration is 1.25, but for simplicity in this paper we refer to the constant $n_{a>0.5\mu m}$ as the INP
concentration. It should also be noted that the parameterization is designed and implemented as
immersion freezing, that is, a pre-existing liquid particle (droplet or drop) is consumed for each
formed ice crystal determined by the parameterization (at the same time, an ice nucleus is
removed from the INP category). An added feature of implementation was to assume that the
largest drops freeze first, followed by the smaller ones over the size spectrum of water drops
when ice nucleation occurs. This implementation yielded the majority of observed large ice
particles, as discussed in FAN2014. This assumption also acknowledges the expectation that the





largest droplets should have a higher probability of containing an INP active at a given
temperature. For contact freezing, we adopt the implementation of Muhlbauer and Lohmann
(2009) for the parameterizations described in Cotton et al. (1986) and Young (1974) to connect
with INP. The contribution from the contact freezing with this parameterization is negligible. As
described in FAN2014, INP is a prognostic variable and over-nucleation is prevented by
applying an upper limit of ice particle concentration.
**2.2 Design of numerical experiments**

In FAN2014, simulations were done for the two nested domains with a horizontal grid-

spacing of 10 and 2 km, respectively. To focus on the orographic clouds over the Sierra Nevada
Mountains and provide a better process-level understanding, we conduct new simulations using a
smaller domain of 300 km × 280 km with a grid-spacing of 1 km (the yellow box in Fig. 1a)
nested within the 2-km grid-spacing domain of FAN2014 (the blue box). The domain grid points
are 301×281 horizontally with 51 vertical levels. The initial and lateral boundary conditions are
produced from the baseline simulations of the 2-km grid-spacing in FAN2014 that were
validated by various observational data.  The lateral boundary data are updated every 3-hours.
The RRTMG shortwave and longwave radiation schemes are used to account for aerosol-cloud-
radiation interactions based on the droplet effective radius calculated by SBM.

CCN in the model is represented by a spectrum with 33 size bins with prognostic CCN

number concentration for each bin. As stated above, the INP denotes the dust/bio particle number
concentration in this region. For the purpose of this study, we conduct sensitivity tests by varying
CCN and INP (i.e., dust/bio particle) concentrations over a wide range from the extremely low to
extremely high concentrations as shown in Table 1. The initial CCN concentrations for the
sensitivity simulations are set to be 30, 100, 300, 1000, and 3000 cm$^{-3}$ (referred to as CCN30,





CCN100, CCN300, CCN1000, and CCN3000 respectively). For each CCN condition,
simulations are conducted with the initial INP concentration of 0.1, 1, 10, and 100 cm$^{-3}$,
respectively, referred to as IN0.1, IN1, IN10, and IN100. Note that 100 cm$^{-3}$ dust/bio means ~100
L$^{-1}$ actual nucleated ice particles at -20$^{o}$C and 1000 L$^{-1}$ at -25$^{o}$C using DeMott et al. (2015). The
vertical profiles of CCN and INP number concentrations at the initial time are uniform below 6
km since observations do not show significant vertical variations as discussed in FAN2014.
Simulations are conducted for both cases, and start at 12:00 pm UTC and run for 12 hours since
the majority of the convective orographic clouds occur during this period. Note the observed
CCN (INP) concentrations for CMOC and WMOC are around 30 (2) and 120 (4) cm$^{-3}$,
respectively.

The CMOC case on FEB16 has cloud top temperatures about 10 degrees colder than the

WMOC case on MAR02, and has higher relative humidity (RH) due to the lower temperature
although the water vapor mixing ratio is much smaller (Fig. 1b-1d). Both cases are under the
influence of atmospheric rivers that provide ample water vapor supply. We note however that the
lower-level wind directions in the two cases are different, with prevailing westerly and
northwesterly on FEB06, and southerly and southwesterly on MAR02.

**3. Results**
**3.1 CMOC – FEB16**
**3.1.1 Precipitation and microphysical processes**

Fig. 2a shows the accumulated surface precipitation averaged over the domain for the

CMOC case (FEB16). Increasing INPs generally enhances the domain-averaged precipitation
except at extremely high CCN concentration (i.e., 3000 cm$^{-3}$), as a result of increased snow



precipitation (Fig. 2c). The sensitivity to INP concentration gets much smaller when INPs are 10
cm$^{-3}$ and larger. Under the low INP condition where the liquid regime is dominant, the
precipitation is first suppressed as CCN increase up to a polluted condition of 1000 cm$^{-3}$ (grey
arrow). This behavior is similar to the CCN effects on shallow warm clouds. As INP is further
increased and mixed-phase clouds are increased, the decreased trend of precipitation with the
increase of CCN is changed to a monotonic increasing trend as shown by the brown arrow in Fig.
2a. The most significant feature of Fig. 2a is the sharp increase of surface precipitation from
CCN of 1000 to 3000 cm$^{-3}$, even at the extremely low INP condition. This is inconsistent with
our previous understanding for deep mixed-phase clouds that precipitation should be
significantly suppressed under the extremely polluted conditions because droplets get too small
to growth efficiently and the riming also becomes very inefficient (Fan et al., 2007; Li et al.,
2008). From Figs. 2b and 2c showing the liquid and snow mass concentrations near the surface
(i.e., at the lowest model level of ~ 40 m above the ground), respectively, we see that (1) snow
dominates the precipitation for the CMOC case and the ratio of warm rain to total precipitation is
very small; and (2) the dramatically enhanced snow explains the sharp increase of precipitation
from CCN of 1000 to 3000 cm$^{-3}$. Note that increasing CCN enhances snow precipitation under
any INP condition (Fig. 2c), and warm rain is totally shut off when CCN are 1000 cm$^{-3}$ or larger
at INP of 0.1 cm$^{-3}$ (Fig. 2b) due to the much smaller sizes of droplets.

By looking at the in-cloud microphysical properties as shown in Fig. 3, increasing CCN

enhances snow number concentration and mass mixing ratio ($N_s$ and $Q_s$, respectively). Especially,
we see a large increase of snow mass from CCN1000 to CCN3000. Cloud ice number
concentration and mass mixing ratio ($N_i$ and $Q_i$, respectively) is also increased. Note ice and
snow are represented with a single size spectrum and a threshold size of 150 μm in radius is used





to separate them. As discussed in Section 2, the major ice nucleation is through the immersion
freezing of DeMott et al. (2015), and with a specification that the largest droplets freeze first
when ice nucleation occurs. Therefore, most of the newly-formed ice particles should be large
and fall into the snow bins, and so $N_s$ and $Q_s$ contribute more significantly to ice number and
mass increase with the increase of CCN than do $N_i$ and $Q_i$. As CCN increases, not only cloud
droplet number concentration ($N_c$) is increased, but also cloud mass mixing ratio ($Q_c$). The large
increase of $Q_c$ when CCN are high, which corresponds to the large increase of $Q_s$, will be
scrutinized a little later. The decrease of raindrop number concentration and mass mixing ratio
($N_r$ and $Q_r$, respectively) is very sharp and warm rain becomes negligible when INP are 1 cm$^{-3}$ or
larger (Fig. 3).

From the process rates of the major microphysical processes shown in Fig. 4, we see that

the increase of $Q_c$ with the increase of CCN and the decrease of $Q_c$ with the increase of INPs are
well explained by the condensation rate (Fig. 4a), although the changes of evaporation have the
same trends as well. As shown in Figs. 4c and 4e, deposition is a more significant process than
riming except in the case of extremely low INP (0.1 cm$^{-3}$) in this CMOC. Increasing CCN
enhances deposition but only enhances riming when CCN are high. The sharp increase of
deposition and riming rates from CCN1000 to CCN3000 explains the sharp increase of snow
with a major contribution from deposition. How deposition and riming are enhanced so
significantly in this case will be elucidated in Section 3.1.2

At the extremely low INPs of 0.1 cm$^{-3}$, the riming rate is similar to the deposition rate in

this CMOC (Figs. 4c and 4e). As INPs increase, the contribution of riming is reduced
significantly because of the reduction of supercooled droplets resulting from increased ice
particles in the mixed-phase zone. Thus, the riming process is liquid-limited in this CMOC. As a



253 result of increased ice particles, deposition is enhanced significantly, and it becomes 3-4 times

254 larger than riming when INPs are 10 cm$^{-3}$. In the observed condition (i.e., CCN are between 30-

255 300 cm$^{-3}$ and INPs range between 1-10 cm$^{-3}$), both deposition and riming contribute to the snow

256 growth but deposition is the major player. When INPs are extremely high (100 cm$^{-3}$), clouds

257 glaciate very fast and liquid droplets that are available for riming are limited, therefore, its

258 contribution is negligible (red line in Fig. 4e).

259  The Wegener–Bergeron–Findeisen (WBF) processes refer to ice depositional growth at

260 the expense of liquid through evaporation in mixed-phase clouds. So the mixed-phase cloud

261 regime where vapor pressure falls between the saturation vapor pressure over water and ice is

262 defined as the WBF regime. As CCN increase, the WBF processes gets stronger as shown in Figs.

263 5a and 5b.  The ratio of the evaporation through WBF to the total evaporation is larger than 0.92

264 in all simulations (Fig. 5a), meaning that drop evaporation in this CMOC occurs predominantly

265 in the WBF regime. There is generally only 50-70% of deposition occurring in the WBF regime

266 even when INP concentration is at 0.1 and 1 cm$^{-3}$ (Fig. 5b), so a significant portion of deposition

267 occurs outside of the WBF regime, and the portion increases as INP increase. Therefore,

268 increasing INPs generally reduces the WBF regime because of the reduced liquid due to

269 enhanced depositional growth. In this CMOC, the ratio of riming occurring in the WBF regime

270 to the total riming is small (generally around 0.2-0.4 in Fig. 5c), meaning that riming mainly

271 occurs outside of the WBF regimes under any CCN and INP conditions. The ratio is increased by

272 CCN but generally decreased by INPs as a result of the increase/decrease of liquid regime,

273 respectively (Fig. 5c).

274  We see that all major microphysical processes (condensation/evaporation,

275 deposition/sublimation, and riming) are highly sensitive to INPs, while generally having much





lower sensitivity to CCN when CCN are below 1000 cm$^{-3}$. The sensitivity of all the major
microphysical processes to CCN gets much more significant when CCN are 1000 cm$^{-3}$ and larger
(Fig. 4), associated with significant changes in dynamics and thermodynamics and will be
discussed in detail below.

**3.1.2 Mechanism of enhanced snow precipitation by highly elevated CCN concentrations**
Since the results of significant enhanced precipitation from CCN1000 to CCN3000 are
unusual, besides verifying the use of identical initial and boundary meteorological conditions in
all the experiments to eliminate simulation differences arising from inadvertent factors, we also
conducted sensitivity tests by restoring the ice nucleation mechanisms to the default
parameterizations (i.e., Meyers et al., 1992 for condensation/deposition and Bigg (1953) for
immersing freezing) in the SBM but this yielded a similar conclusion. So, the significantly
increased snow precipitation associated with elevated CCN concentrations is not the result of the
particular ice-forming parameterization or the implementation approach of the parameterization.
Since the precipitation enhancement begins at 1400 UTC, which is a couple of hours into
the simulations, we focus on the time period of 14-1600 UTC and use the simulations of
different CCN concentrations with INP concentrations of 1 cm$^{-3}$ to examine the mechanism. By
taking a close look at ice nucleation (using model outputs every 6 min), we find that the total
nucleated ice particle number concentration is increased as CCN increase and there is a large
jump from CCN1000 to CCN3000 (Fig. 6a). The increase is caused by more cloud points that
have ice nucleation occurring (Fig. 6b) and the enhanced nucleation rate (i.e., the nucleated ice
particles per liter air volume within a hour) in the lower altitudes (Fig. 6c). Considering that the
major ice formation mechanism is immersion freezing in this study, which requires the existence





of drops for primary nucleation of ice, it means that there is much more supercooled liquid cloud
area/volume available for nucleation in the lower altitudes as CCN increase (Fig. 6e). As shown
in Fig. 6d, the increase of cloud water ($Q_c$) that is supercooled, since the warmest cloud
temperature is below 0°C in this case, is very significant, with a big jump from CCN1000 to
CCN3000, corresponding to the large increase of snow precipitation. From CCN1000 to
CCN3000, the increase of the supercooled liquid area is especially drastic (Fig. 6e).

What causes the drastic increase of $Q_c$ and a more widespread supercooled liquid cloud

regime that is available for ice nucleation? We know that the increased drop surface area with the
increased CCN can increase condensation, but it cannot explain such a drastic increase of the
condensation rate averaged over the entire domain as shown in Fig. 6f. We find that over the
domain the updraft area (i.e., grid points with $w > 1$ m s$^{-1}$) is increased significantly with CCN
with a jump from CCN1000 to CCN3000 as well (Fig. 7a), but the averaged updraft velocity
does not change significantly (Fig. 7b), suggesting that much more convection occurs to form
more clouds in the domain as CCN increase, especially in CCN3000. From the spatial
distribution, we see that the increase of clouds is the most prominent around the valley and
foothills (i.e., the lower-part of the windward slope of the Mountains). The cross sections of
cloud water, rain and ice/snow mass mixing ratios at 1400 UTC clearly show that more clouds
form over the valley and foothills in CCN3000, while in CCN30 clouds over the valley are fewer
and clouds are shallower over the valley and foothills (Fig. 8a). we see much more invigorated
mixed-phase clouds in CCN3000 compared with CCN30. The mixed-phase clouds start from the
foothills in CCN3000 (Fig. 8c), while CCN30 does not have the mixed-phase clouds present
until the regions above the middle and upper part of the mountain slope. This explains the
increased ice nucleation rate at the lower altitudes as shown in Fig. 6c.



Changes of cloud fields described above must involve dynamic and thermodynamic
changes. By examining the differences of dynamic and thermodynamic fields between CCN3000
and CCN30 (Fig. 9), we clearly see that a band of increased water vapor and relative humidity
(RH) from the valley/foothills to the mountain at the higher altitudes (Fig. 9a-b). The
corresponding temperature is only slightly decreased (Fig. 9c), which should not much affect the
saturation water pressure and ice nucleation efficiency by much. So, the increased RH is mainly
caused by the increased water vapor, and this increase can be up to 8% in RH (e.g., from RH of
70% to 78%). The large increase of $Q_v$ and RH is mainly a result of changed local circulation as
shown in Figs. 9d-e: the wind blowing to windward slope (zonal wind) gets stronger from
CCN30 to CCN3000 (within ~ 2 km above the ground) over the slope. In the cases of
atmospheric rivers, the stronger zonal wind transport means an increase of moisture transport to
the Mountains.
The changes of winds are only significant at the slope of the Mountains and occur only
after 2 h of the simulations (Fig. 10a), suggesting that it stems from more latent heat release as a
result of more clouds over the valley and foothills (feedbacks of radiation and precipitation take
much longer time especially considering the two-hour time is 4- 6 am LST). The clouds at the
valley/foothills locations are generally shallow. Many literature studies, including both
observations and model simulations, have shown that CCN enhance shallow cloud formation and
deepen shallow clouds (e.g., Chen et al. 2015; Yuan et al. 2011; Pincus and Baker 1994; Koren
et al. 2014), which can be due to various reasons such as cloud lifetime effect, enhanced
turbulent convection by larger entrainment rates as a result of stronger evaporation, and greater
latent heat release due to larger drop surface area for stronger condensation. We find that
condensation is indeed much enhanced over the valley/foothills from CCN30 to CCN3000 with





INP of 1 cm$^{-3}$ (Fig. 9f), which results in much reduced supersaturation with respect to water
(supersaturation around the cloud base in CCN30 at 1300 UTC is about 0.28% while only 0.04%
in CCN3000). The enhanced condensation as well as the cloud lifetime effect (i.e., conversion of
smaller droplets into rain is slow and cloud can be sustained for a longer time) contributes to
more shallow clouds at the valley/foothills. The more latent heat resulting from enhanced
condensation leads to the change of local circulation, which transports more moisture to the
windward slope of the Mountain,resulting in more active mixed-phase clouds and snow
precipitation through enhanced deposition and riming. In addition, over the Mountains more
supercooled liquid would be lifted to the higher altitudes in the polluted condition, forming
ice/snow more efficiently through immersion freezing at the colder temperature, which
contributes to more snow precipitation as well.

It should be noted that the mixed-phase clouds over the Mountains are the key to the

enhanced precipitation by CCN. In the sensitivity tests based on the WMOC case where ice-
related microphysics is turned-off (the WMOC case is chosen because ice processes are weak),
precipitation is dramatically suppressed from CCN of 30 cm$^{-3}$ to 3000 cm$^{-3}$ (Fig. 11a) and there
is almost no precipitation at the valley and windward slope in CCN3000 due to extremely small
droplets. However, we still see the change of the local circulation over the slope as a result of
enhanced condensation (Fig. 11b). Therefore, presence of ice is a necessary condition for such a
large increase precipitation by CCN. Without ice processes (e.g., under the warm season with
warm clouds only), precipitation over the Mountains can not form efficiently in such a polluted
condition even with the increased moisture. But the added latent heat from condensation of vapor
to water is still the main energy source of the invigoration.

In summary, increasing CCN forms more clouds at the valley and foothills (generally





shallow) through much enhanced condensation, which induces a local circulation change due to
more latent heat release that enhances the zonal transport of moisture, leading to the invigoration
of the orographic mixed-phase clouds and drastically increased snow precipitation in this CMOC
case. Therefore, aerosol impacts on orographic mixed-phase clouds can be extraordinary in
extremely polluted conditions, especially under the influence of atmospheric rivers. Besides the
the key role of ice processes for leading to greatly enhanced precipitation, orographic dynamics
is another important factor since we do not see such impacts in the sensitivity tests where the
terrain height is set to be 600 m for the locations with a terrain height > 600 m (precipitation
becomes very small in those sensitivity tests and the increase from CCN30 to CCN3000 is small
as well).

The increases of $Q_v$ and RH are the most significant from CCN1000 to CCN3000 due to

non-linearity of aerosol-cloud interactions, explaining the large increase of snow precipitation. It
is worth noting in CCN3000, warm rain is completely shut off (left column in Fig. 8b), therefore,
much more cloud water can be transported to higher altitudes for more immersion freezing,
which further enhances the snow precipitation. This likely contributes to the steep increase in
precipitation when CCN reach 3000 cm$^{-3}$.

### 3.1.3 Supercooled water content (SCW) and cloud phase

Through changing the microphysical process rates, CCN and INP impact the cloud

phases and supercooled water content (SCW). Fig. 12 shows that INPs have the most striking
impact on SCW. Increasing INPs enhances ice particle formation, and then facilitates the
deposition and riming processes in this CMOC as discussed in Section 3.1.1. The enhanced
deposition in the WBF regime, along with riming, leads to a faster conversion of liquid to ice in



the mixed-phase and glaciates the clouds faster. Therefore, SCW is substantially reduced as INPs
increase (Fig. 12a). For example, in the case of CCN300, a significant amount of liquid mass
fraction (0.1) exists at the temperature of -30°C when INP is 0.1 cm$^{-3}$. Such temperature is
increased to -20, and -10°C as INPs are increased to 1 and 10 cm$^{-3}$, respectively. In the extremely
high INP case (100 cm$^{-3}$), there is nearly no supercooled water. As a result, the fractions of cloud
phases are dramatically changed (Fig. 13a). As expected, higher INP concentrations decreases
the fractions of liquid and mixed phases due to increasing the fraction of ice phase. In this
CMOC, the cloud phases are most sensitive to INPs at relatively low concentrations. For
example, when INPs increase from 0.1 to 1 cm$^{-3}$, which is likely common for this region based
on observations in the past field campaigns, the liquid phase fraction is reduced by nearly half
and the ice phase fraction is increased by 2 times or larger (Fig. 13a). Note that the effects of
INPs on cloud phase and SCW presented in this study may represent the upper limit because ice
formation is mainly through immersion freezing that transforms the large liquid particles to ice
particles when ice forms.
Compared with the effects of INPs, the magnitudes of CCN effects on SCW and cloud
phases are much smaller but still significant (the lines with same color but different line styles in
Fig. 12). Moreover, the sign is opposite. Increasing CCN generally increases SCW slightly (Figs.
12a). The impact of CCN on cloud phases is generally small, except when INPs are very low, i.e.,
at 0.1 cm$^{-3}$ (Figure 13a). In this low INP case, increasing CCN increases ice phase fractions and
reduces the mixed-phase fraction when CCN are relatively low. This is because liquid clouds are
dominant so such clouds are sensitive to the CCN-enhanced ice nucleation as discussed in the
section 3.1.2.




**3.2 WMOC – MAR02**

For this warm mixed-phase cloud case, the surface accumulated precipitation is

suppressed by increasing CCN when CCN are lower than 1000 $cm^{-3}$ (Fig. 14a), which is
different from the case of CMOC where the sign of CCN impact on precipitation depends on INP
concentration. This is because the clouds in this WMOC behave similarly as warm clouds due to
less efficient ice nucleation at the warm cloud temperatures. When CCN are lower than 1000 $cm^{-}$
$^{3}$, the large decrease of warm rain (Fig. 14b) overpowers the slight changes of snow precipitation
(Fig. 14c). Similar to the CMOC case, we see a drastic increase of surface precipitation from
CCN1000 to CCN3000, also due to drastic increase of snow precipitation. Increasing INPs
enhances surface precipitation in a more significant manner than that in CMOC. In other words,
the WMOC is more sensitive to INPs than the CMOC.

The in-cloud microphysical properties also show similar results as for the CMOC: the

steep increases of the snow mass and cloud water mixing ratios from CCN1000 to CCN3000
(Fig. 15). We have done the same investigation as in Section 3.1.1, and found the mechanism
causing the increased cloud water and the snow production is similar as that in CMOC, that is,
increasing CCN forms more shallow clouds at the large area of valley and foothills, which
induces a change of local circulation significantly through more latent heat release, which in turn
increases the zonal transport of moisture to the windward slope of the mountains. Additionally,
more abundant warm rain is present at the wide valley area in this case when CCN is low (30 $cm^{-}$
$^{3}$) compared with the CMOC. The suppression of warm rain as CCN increase is very significant
as shown in Figs. 14b and 15. Over the Mountain, this suppression increases $Q_c$ and allows more
cloud water to be transported to the higher altitudes along the slope where immersion freezing is
able to occur at lower temperatures. Ice multiplication through the Hallet-Mossop



parameterization (Hallet and Mossop, 1974) in this WMOC contributes to ice particle
concentration by 10-15% when CCN are 30 cm$^{-3}$ and INPs are 1 cm$^{-3}$ in our model simulation
with the fast version of SBM in which ice habits are not considered. Therefore, as more ice
particles form from immersion freezing when CCN increase, the ice multiplication processes
would further increase ice crystal formation although the contribution is relatively small in the
model simulation. Past observation studies suggested that ice multiplication through rime-
spintering does occur in the orographic mixed-phase clouds of this region (Marwitz 1987;
Rauber 1992). We do not yet have a clear understanding of the importance of this process in
contributing to ice formation in reality. After more ice particles form, the subsequent ice
depositional and riming growth processes form efficient snow precipitation. The CCN impact on
local circulation change is more significant in this case compared with the CMOC, probably due
to much more shallow warm clouds in the valley.

Different from the CMOC case, riming is a more efficient ice growth process to form

snow than deposition in this case except when INP concentrations are extremely high (100 cm$^{-3}$)
where both riming and deposition contribute in a similar magnitude (Fig. 16). In addition, the
riming rate is increased as INP concentrations increase, which is opposite to that of CMOC. This
is because the WMOC is ice-limited and there are not enough ice particles to collide with liquid
particles when INP numbers are low, therefore, increasing INPs boosts ice particles and allows
more riming to occur. In contrast, the CMOC case is liquid-limited, so increasing INPs reduces
liquid particles available for riming due to ice depositional growth. We also see that
condensation and evaporation rates are generally more than 2 times larger in this case compared
with CMOC and both rates increase more significantly with CCN concentration in this WMOC.
This is related to the dominance of liquid clouds in the WMOC.  The more significant increase of





condensation by increasing CCN compared with the CMOC is likely a result of the more
significant change of the local circulation that is associated with more shallow clouds forming at
the valley. Increasing INP number concentrations reduces evaporation simply because of the
reduction of liquid due to the increased deposition and riming.

Similarly as in the CMOC, increasing CCN enhances the WBF process for this WMOC

as more droplet evaporation and ice deposition occur (Figs. 17a and 17b). With the increase of
CCN, the domain-mean riming rate is not changed much until CCN of 1000 cm$^{-3}$ (Fig. 16e), but
the riming rate in the WBF regime is increased (Fig. 17c), possibly due to larger ice particles
resulting from stronger deposition growth in the WBF regime.

Similar results regarding CCN and INP impact on supercoooled water content are

obtained in the WMOC case as in the CMOC case: increasing INPs dramatically reduces SCW
and increases cloud glaciation temperature, while increasing CCN has the opposite effect with
much smaller significance (Fig. 12b). Compared with the CMOC, the effects of INP on SCW are
a little smaller but CCN effects are a little larger. The liquid phase fraction (number fraction of
cloudy grid points for which the liquid represents 99% or more of the condensate mass)
decreases significantly as INPs increase (Fig. 13b). Correspondingly the fractions of the mixed-
phase and ice phase cloud volumes increase due to increased ice nucleation. Similar to the
increased riming as INPs increase, the mixed-phase fraction is increased as well in the WMOC,
which is opposite to the case for CMOC, as a result of the ice-limited condition in the WMOC
versus the liquid-limited condition in the CMOC. Note that INP effects are more significant at
higher INP concentrations in this case, while in CMOC the sensitivity decreases as INP increases,
suggesting that the optimal INP concentration for the maximum INP impact is higher in warmer
clouds than the colder clouds, because of less efficient ice formation at the warmer cloud



temperatures. The CCN impacts on cloud phase are more significant in this WMOC compared
with those in CMOC. The decreased liquid cloud fraction with the increase of CCN is a
consequence of the large increase of ice phase fraction resulting from more active cold-cloud
processes, since the total cloud fraction sums up to 1 (Fig. 13b).

**4. Conclusions and Discussion**

Extending the previous study of Fan et al. (2014), we conducted new simulations at

higher resolution and further sensitivity studies based around the same two mixed-phase
orographic clouds forming on the Sierra Nevada barrier under the influence of atmospheric rivers
that were our focus from the CalWater 2011 field campaign, to quantify the response of
precipitation to changes of CCN and INP and to examine CCN and INP impacts on SCW and
cloud phases. The two mixed-phase cloud cases have contrasting thermodynamics and dynamics:
FEB16 has cold cloud temperatures and northwesterly wind flow at lower-levels (i.e., CMOC),
while MAR02 has about 10 °C warmer cloud temperatures and southerly wind flow (i.e.,
WMOC).

It is found that, in the CMOC case, deposition contributes more significantly to snow

production than the riming because deposition process is efficient at the cold cloud temperatures
(from -22 to -32 °C) in this case. In the WMOC, riming generally contributes more significantly
because the deposition growth process is less efficient at the warmer temperatures (generally
warmer than -20 °C in this case), except in the extremely high INP case where both riming and
deposition contribute similarly.

We find that increasing INP concentrations enhances snow precipitation on the windward

slope of the Sierra Nevada Mountains in both CMOC and WMOC cases. With the increase of



INPs, the increased ice nucleation via immersion freezing enhances snow formation by
intensifying depositional growth of ice in the CMOC while both deposition and riming
contribute in the WMOC. Increasing INPs reduces riming in the CMOC, because of the liquid-
limited condition in which more efficient depositional growth at higher INP number
concentrations glaciates clouds and reduces liquid particles available for riming. However, in the
ice-limited conditions of WMOC, increasing INPs boosts ice particle concentrations so that more
riming can occur in a liquid-rich condition. For the same reason, increasing INPs suppresses the
WBF processes due to reduced liquid particles.
The CCN impacts on precipitation are complicated, depending on cloud temperature, and
concentrations of CCN and INP. When CCN are lower than 1000 cm$^{-3}$, boosting CCN
concentrations slightly increases snow precipitation, but the total precipitation can be increased
or decreased depending on INP concentrations in the CMOC. In contrast, in the WMOC,
increasing CCN suppresses the total precipitation due to the large suppression of warm rain
production. We find a drastic increase of snow precipitation by increasing CCN when CCN are
high (1000 cm$^{-3}$ or larger), consistently in both CMOC and WMOC, as a result of increased
deposition and riming rates. The mechanism by which this occurs is through increasing CCN
forming more shallow clouds at the wide valley area and foothills, which induces a change of
local circulation through more latent heat release and increases the zonal transport of moisture to
the windward slope of the Mountains. This results in much more invigorated mixed-phase clouds
with enhanced deposition and riming processes and therefore much more snow precipitation.
Additionally, over the mountains, the suppression of warm rain as CCN increase allows more
cloud droplets to be transported to the higher altitudes where immersion freezing is able to occur
efficiently, contributing to the enhanced snow as well. This effect is most significant when warm





rain is completely shut off at CCN of 1000 cm$^{-3}$ and higher.

Increasing INP concentrations dramatically reduces supercooled water content and

increases cloud glaciation temperature, while increasing CCN has the opposite effect but with
much smaller significance. As expected, the fraction of liquid phase clouds is decreased and the
ice phase fraction is increased by increasing INP in both cases. However, we see a decreased
fraction of mixed-phase clouds by INP in the CMOC but increased in the WMOC, relating to the
liquid-limited condition in the former where increasing ice formation enhances cloud glaciation,
while the ice-limited condition in the latter in which more liquid clouds are converted to mixed-
phase clouds as INPs increase. Compared with the effects of INPs, the magnitudes of CCN
effects on SCW and cloud phases are much smaller and the signs are opposite. Increasing CCN
generally enhances SCW in both cases. The relative fractions of cloud phases are not much
impacted by CCN in the CMOC, except when INP is very low (i.e., 0.1 cm$^{-3}$). However, in the
WMOC, increasing CCN evidently decreases liquid cloud fraction but increases ice phase
fraction. Thus, cloud phases in the WMOC have a large sensitivity to CCN compared with
CMOC.

This study provides a better understanding of the CCN and INP effects on orographic

mixed-phase cloud properties and precipitation. The result that CCN dramatically increase snow
precipitation over the mountains when CCN are high (1000 cm$^{-3}$ or larger) as a result of modified
cloud properties at the valley and foothills is different from previous modeling studies in the
literature. The mechanism for the drastic increase of the snow precipitation by CCN at the very
polluted condition is new, and it suggests a strong impact of the shallow clouds at the valley and
foothills on the mixed-phase clouds and precipitation over the mountains. It is worthy noting that
we do not see such a significantly increased precipitation by CCN in the sensitivity tests without





ice-related processes or without topography, suggesting that ice processes in the mixed-phase
clouds and orographically-forced dynamics are the key factors for such CCN effects.
Over the region of Sierra Nevada Mountains, CCN of above 1000 cm$^{-3}$ would be an
extreme condition. Therefore, this mechanism would not occur usually and the change of
precipitation would not be much when CCN is less than 1000 cm$^{-3}$ as shown in Fig. 2a and 14a
in the normal conditions over this region. It is shown precipitation suppression by CCN in the
relatively warm situations, in agreement with the observations of Rosenfeld and Givati (2006).
However, for many polluted regions such as China and India where CCN of above 1000 cm$^{-3}$ are
quite common, this mechanism may have very important implications for orographic
precipitation extremes and water cycles.
It should be noted that the results of CCN and INP impacts on the precipitation and
supercooled water content may represent an upper limit since the major ice nucleation in the
simulations is through immersion freezing that converts largest liquid drops into ice or snow
directly when ice nucleation occurs, leading to very efficient conversion of liquid to ice/snow
and then strong ice growth processes to form snow.
In our study, we do not see significant spillover effect of snowfall (i.e., decrease at the
windward slope and increase at the leeside slope by increasing CCN) as found in Saleeby et al.
(2011). Precipitation mainly forms on the windward slope of the Sierra Nevada Mountains and
the increase of the snow precipitation is more significant on the windward slope than on the lee
side in both cases. The different results between our study and Saleeby et al. (2011) could be
related to different locations of the clouds over the mountain and/or different mountain
topography, or the presence of a low-level barrier jet in the atmospheric river environment that
reduces the cross barrier flow.




**Acknowledgements**

This study was supported by the California Energy Commission (CEC) and the Office of Science
of the U.S. Department of Energy as part of the Regional and Global Climate Modeling program.
PNNL is operated for DOE by Battelle Memorial Institute under Contract DE-AC06-
76RLO1830.



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





Table 1 Model simulations that are run for different CCN and dust concentrations. Please note
that INP denotes dust/bio particle number concentration with particle size > 0.5 μm for use in the
parameterization of DeMott et al. (2015), as described in FAN2014.

|  |  | INP (cm$^{-3}$) | | | |
| --- | --- | --- | --- | --- | --- |
|  |  | 0.1 | 1 | 10 | 100 |
| | 30 | x | x | x | x |
| | 100 | x | x | x | x |
| CCN (cm$^{-3}$) | 300 | x | x | x | x |
| | 1000 | x | x | x | x |
| | 3000 | x | x | x | x |





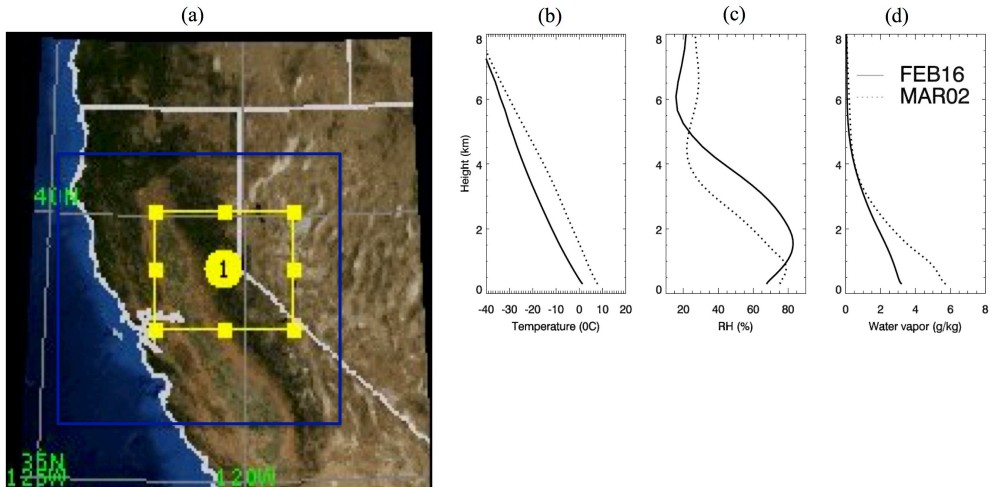

Fig. 1 (a) The simulation domain (yellow box), and the vertical profiles of (b) the
temperature, (c) RH, and (d) water vapor for CMOC (FEB16) and WMOC (MAR02).
(b)-(d) are domain mean values during the model simulation time period. The blue
box in (a) denotes the domain of 2 km resolution simulations done in FAN2014.




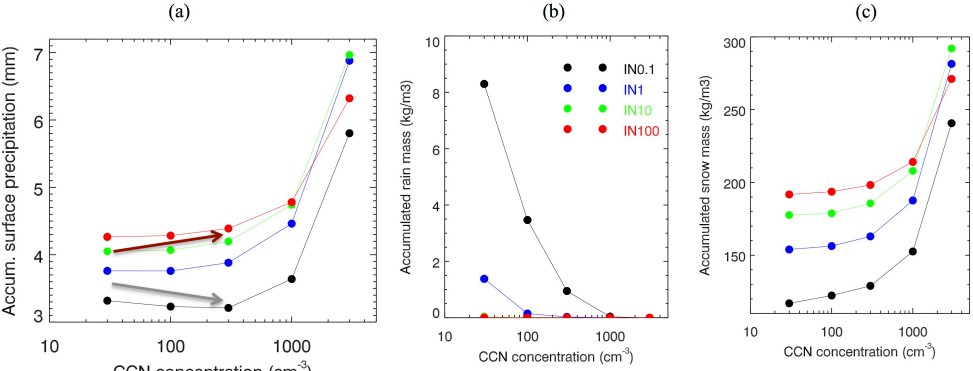

Fig. 2 (a) The domain-mean accumulated surface precipitation, and the accumulated (b) rain and (c) snow mass concentrations at the lowest model level (~ 40 m above the surface) during the simulation time period for CMOC. All domain- mean calculation excludes the lateral boundary grid points in this study.




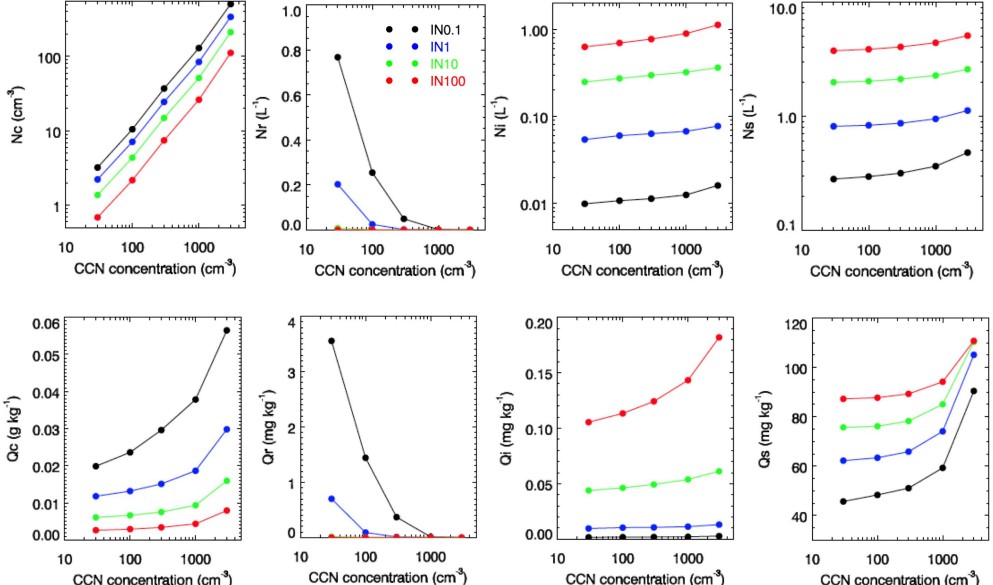

Fig. 3 The number concentrations (top row) and mass mixing ratios (bottom row) of
droplet (1st column), rain (2nd column), cloud ice (3nd column), and snow (4th
column) for CMOC. The data are averaged over the grid points over the domain by
excluding the lateral boundary grid points below the 7 km altitude and over the
simulation time by excluding the initial two hours.



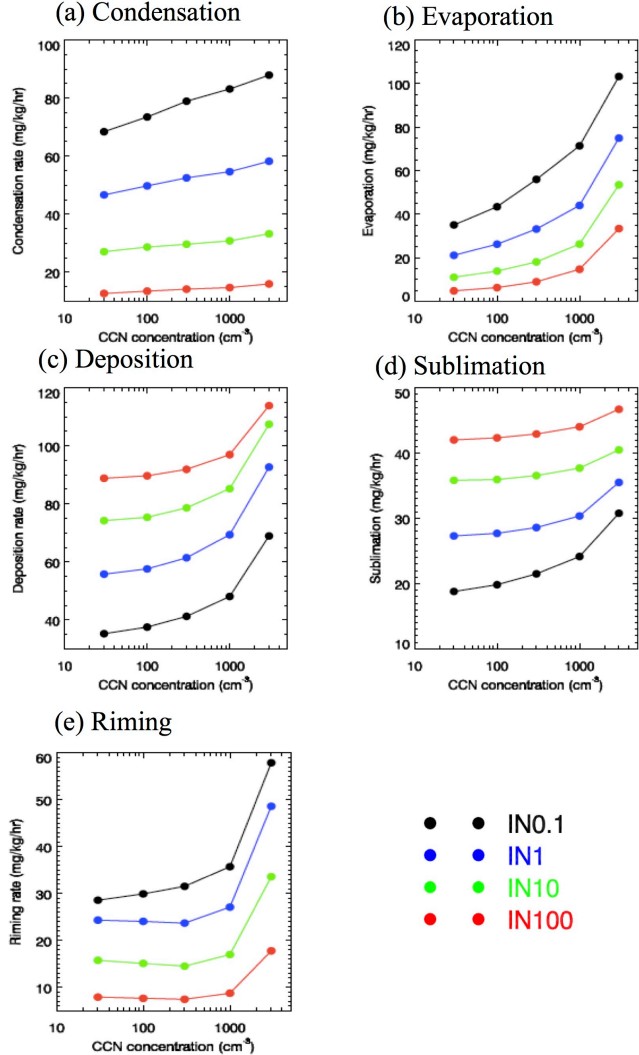

Fig. 4 The microphysical process rates of (a) condensation, (b) evaporation, (c) deposition, (d) sublimation, and (e) riming for CMOC. The model outputs for the process rates are in every 6 min frequency, and the data shown in the plots were processed in the same way as Fig. 3.





(a)        (b)        (c)

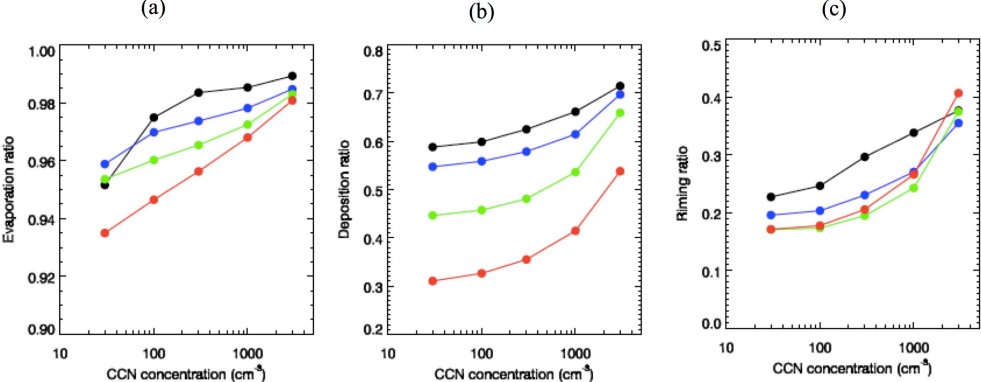

Fig. 5 (a) The ratio of evaporation occurring in the WBF regime that is defined as the grid points where the WBF processes occur) to the total evaporation for the CMOC case. (b) and (c) are the same as (a), except for deposition and riming, respectively. Data were processed in the same way as Fig. 3.



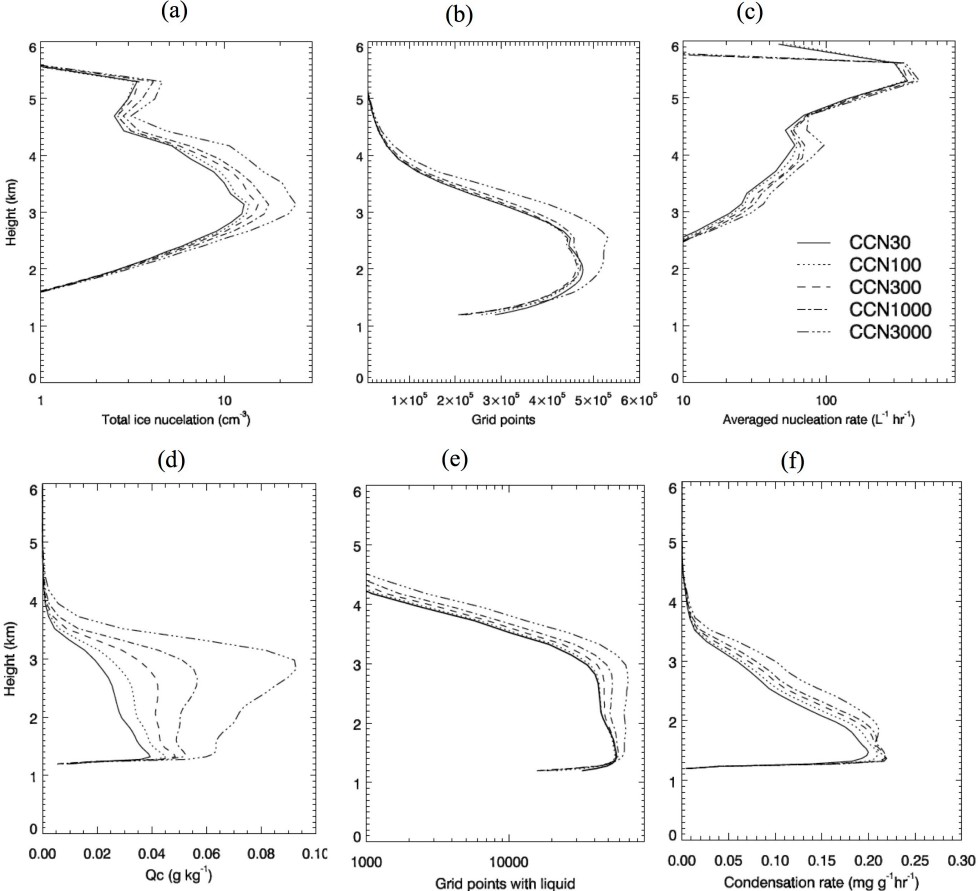

Fig. 6 Vertical profiles of (a) total nucleated ice particles, (b) the total grid points where ice nucleation occurs, (c) the ice nucleation rate averaged over the total ice nucleation grid points, (d) domain-mean cloud water content ($Q_c$), (e) the total grid points that have liquid (i.e., the liquid water mixing ratio is larger than 1.e-5 kg kg$^{-1}$), and (f) the domain-mean condensate rate during 1400-1600 UTC for the CMOC case.




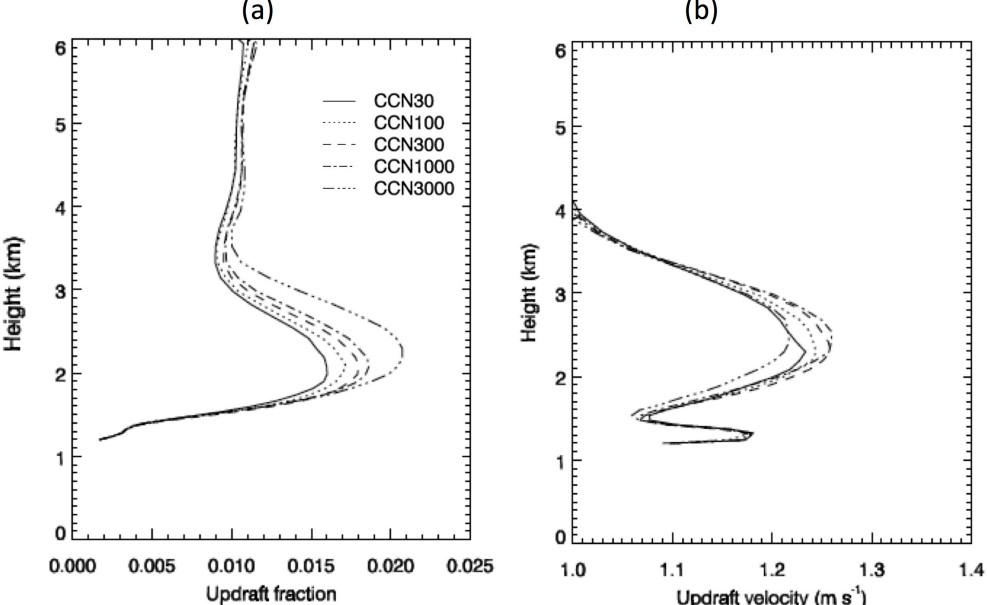

Fig. 7 (a) The fraction of updraft grid points with vertical velocity larger than 1 m s$^{-1}$ relative to the total domain grid points, and (b) the mean updraft velocity for the grid points larger than 1 m s$^{-1}$ over 1400-1600 UTC for the CMOC case.





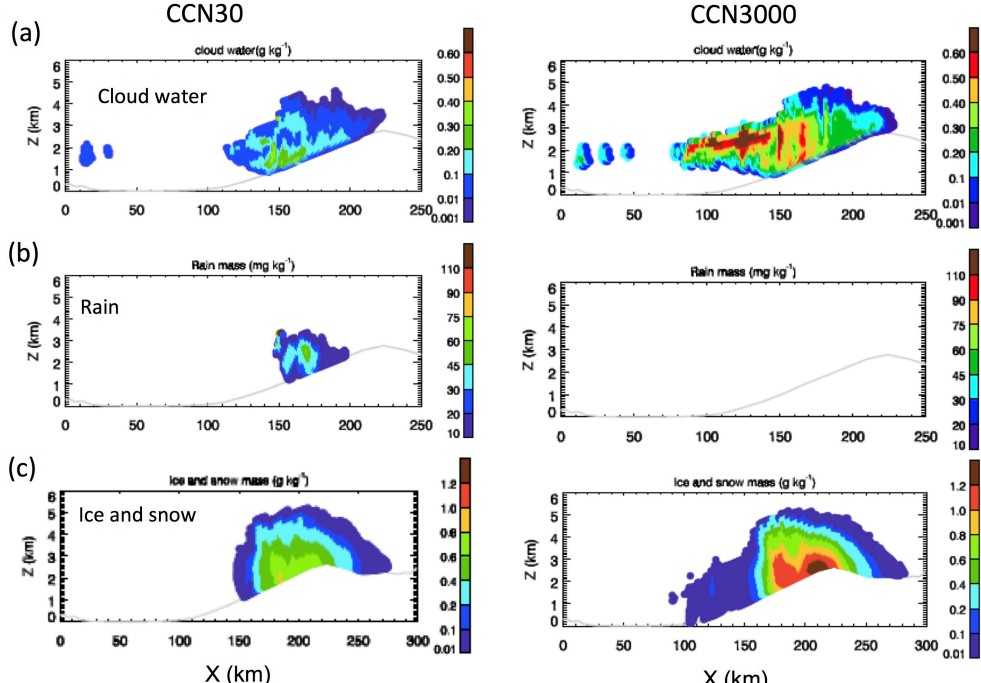

Fig. 8 The west-east cross section of (a) cloud water content, (b) rain water content, and (c) ice and snow water content for CCN30 (left) and CCN3000 (right) with INP of 1 cm$^{-3}$ at 1400 UTC averaged over the 20 km wide area zonally for the CMOC.



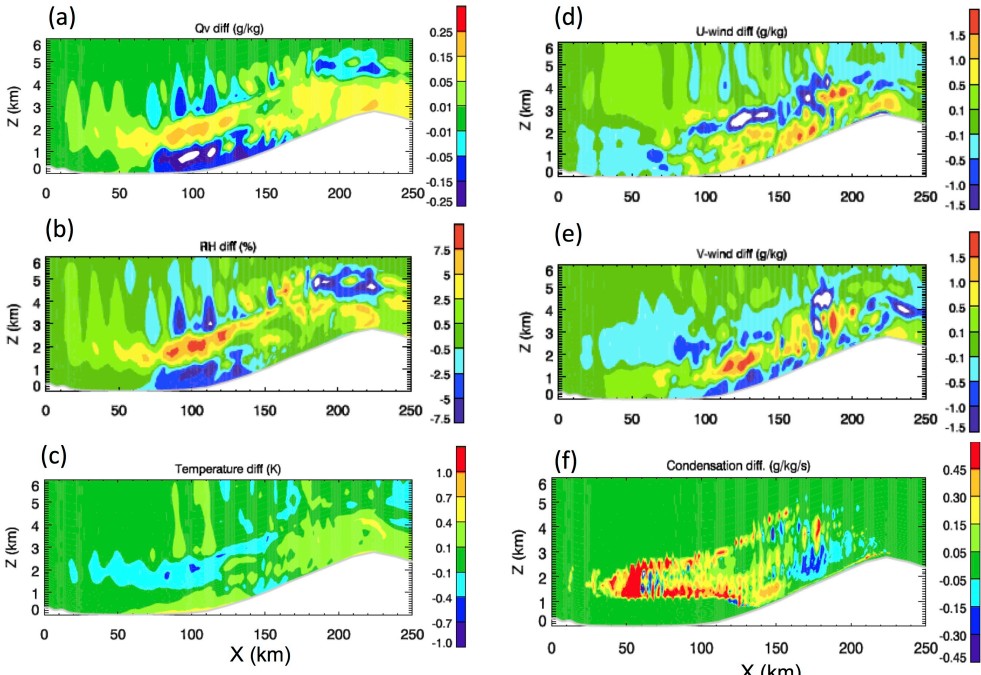

Fig. 9 Differences of (a) water vapor, (b) RH, (c) temperature, (d) U-component of the wind, (f) V- component of the wind, and (f) condensation rate between CCN3000 and CCN30 with INP of 1 cm$^{-3}$ for the CMOC. The cross section area is same as Fig. 8. The time is at 1400 UTC except that the condensation rate used for the difference calculation is the sum of that from 1300-1400 UTC to show an accumulated value over 1-hour period before 1400 UTC.





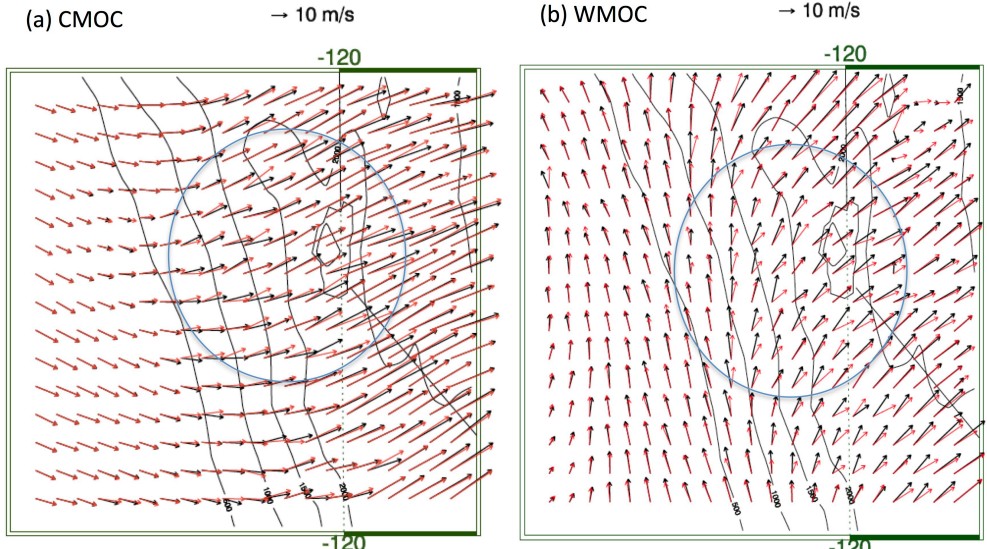

Fig. 10 The spatial distribution of wind field at about 1.7 km above the ground for (a) CMOC and (b) WMOC at 1400 UTC. The red color denotes CCN3000 and black color denotes CCN30 with INP of 1 cm$^{-3}$. The blue cycle is to mark the area with significant changes of wind (i.e., over the wind ward slope of the Mountain).





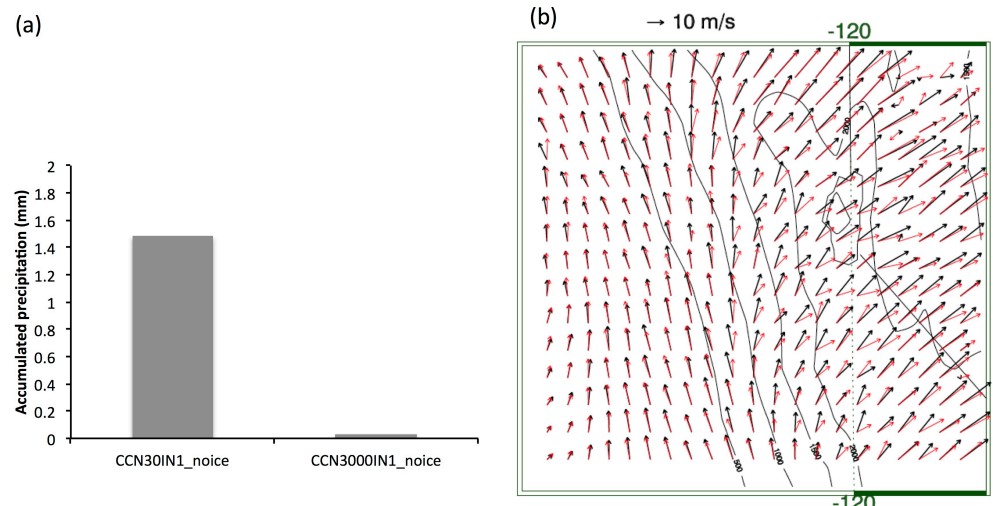

Fig. 11 Results for the two simulations without ice-related microphysics, i.e., CCN30IN1_noice and CCN3000IN1_noice, which are based on CCN30IN1 and CCN3000IN1, respectively, for the WMOC case: (a) the domain averaged accumulated precipitation, and (b) the spatial distribution of wind field at about 1.7 km above the ground at 1400 UTC. The red color on (b) denotes CCN3000IN1_noice and black color denotes CCN30IN1_noice.





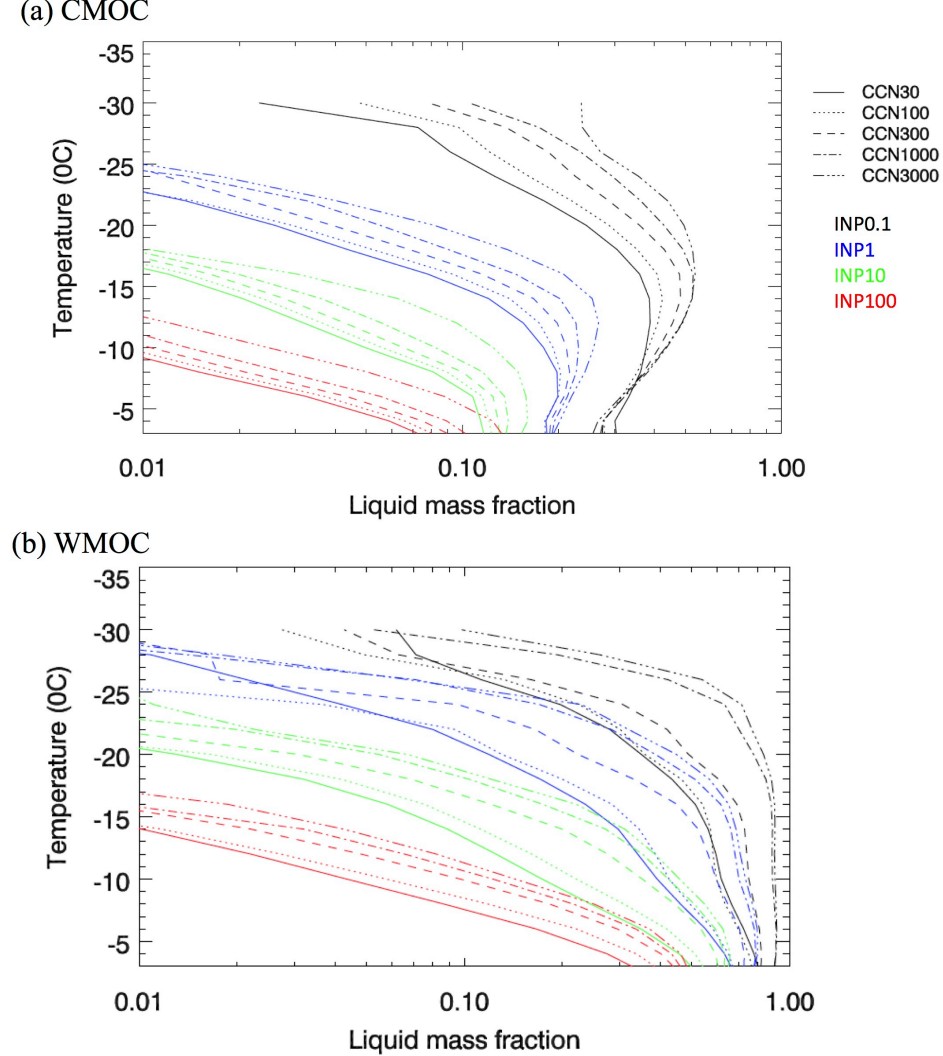

Fig. 12 The liquid mass fraction vs. temperature for the (a) CMOC and (b) WMOC over the simulation time by excluding the initial two hours. The liquid mass fraction is calculated for each temperature bin of a 2 K interval based on the total liquid water mixing ratio (droplets + raindrops) divided by the total condensate mixing ratio. The different line styles denote different CCN concentrations and different colors denote different INP concentrations.




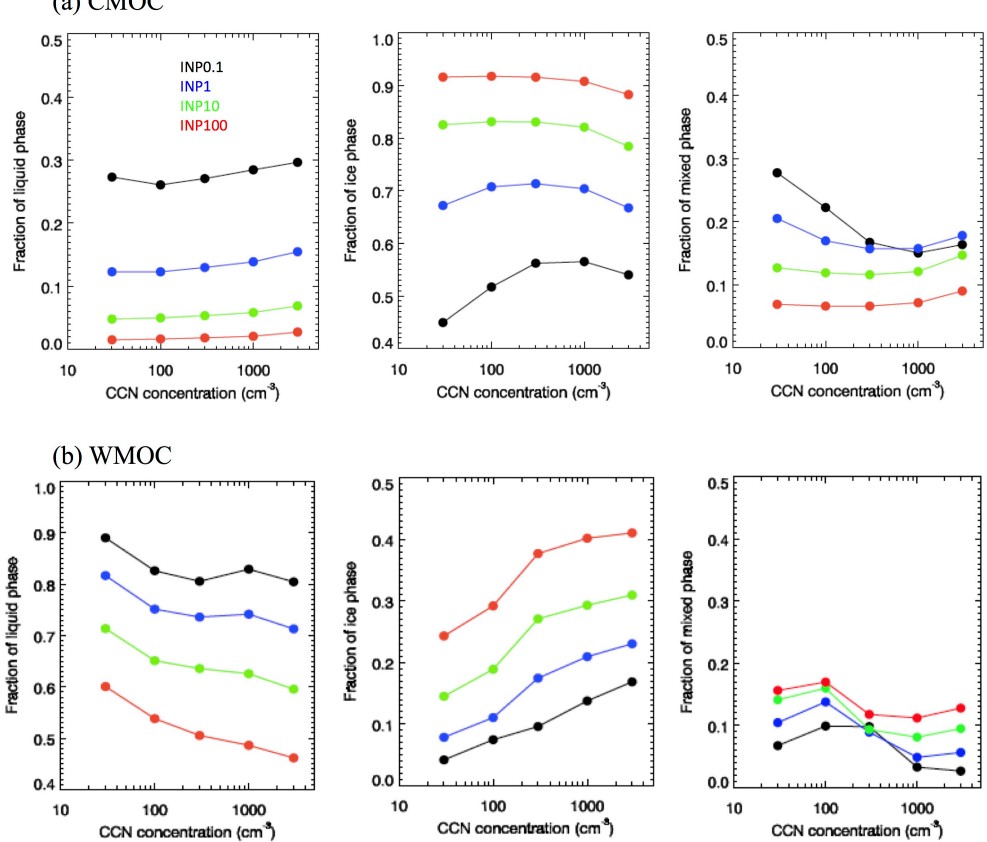

Fig. 13 The fraction of the liquid phase (left), ice phase (middle), and mixed-phase (right) for the (a) CMOC and (b) WMOC over the simulation period by excluding the initial two hours. The cloud phase for each cloud grid point that has a total condensate mass of larger than $1 \times 10^{-5}$ kg kg$^{-1}$ is identified based on the ratio of liquid to ice water mixing ratios. If the ratio is larger than 0.99 or smaller than 0.01, the grid point is identified as liquid phase or ice phase, respectively. Between these values is identified as mixed-phase. The fraction for each cloud phase is calculated by the number of grid points identified for the phase divided by the total number of the grid points of all three phases. So, the fractions of all three add up to 1 for each simulation case.




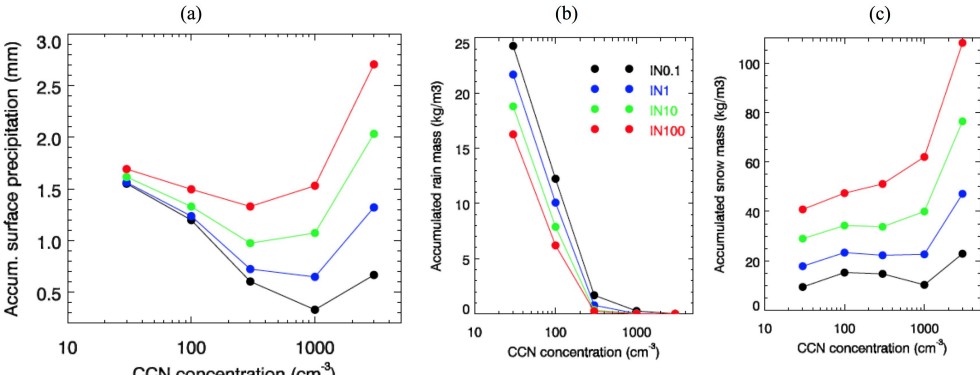

Fig. 14 Same as Fig. 2, except for the WMOC case.



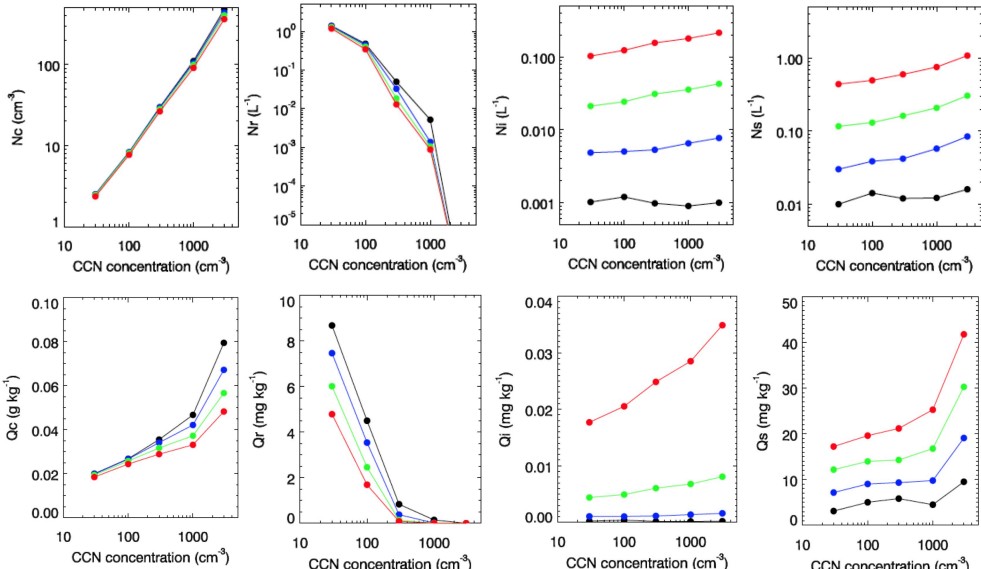

Fig. 15 Same as Fig. 3, except for the WMOC.



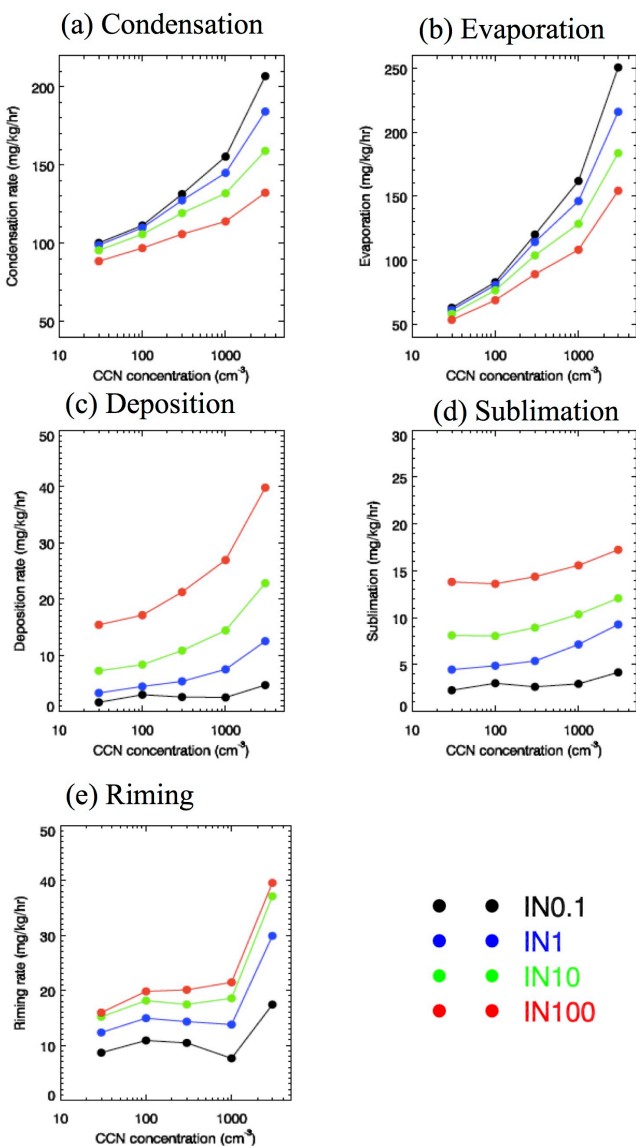

Fig. 16 Same as Fig. 4, except for the WMOC.





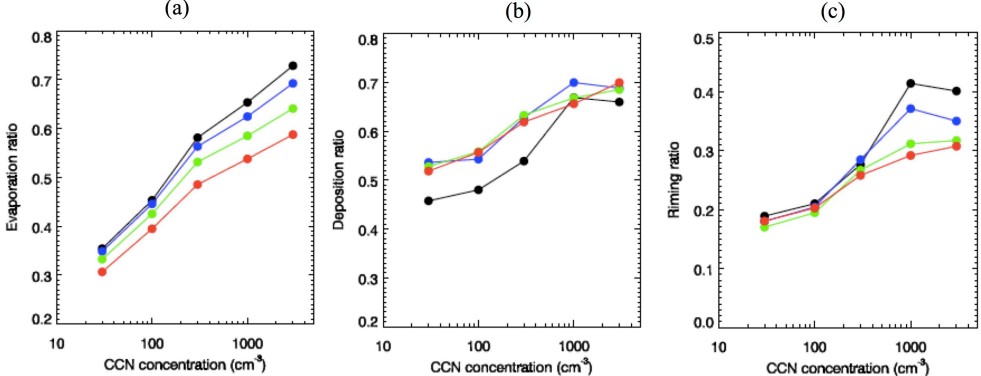

Fig. 17 Same as Fig. 5, except for the WMOC.