# Peer review of "Effects of Cloud Condensation Nuclei and Ice Nucleating Particles on Precipitation Processes and Supercooled Liquid in Mixed-phase Orographic Clouds Jiwen Fan1\*, L. Ruby Leung1, Daniel Rosenfeld2, Paul J. DeMott3 1 Atmospheric Science & Global"

_Atmospheric Chemistry and Physics, 2016_

## Referee Comment (RC1) · Anonymous Referee #1 · 12 Oct 2016

Fan et al. report on an aerosol-cloud-precipitation process modeling study regarding two cases from CalWater 2011. The advantage of this work over FAN2014 is based on the comparison of variable cloud phase conditions (WMOC versus CMOC), providing an added level of detail. One of the more surprising findings is the increase in snow precipitation when CCN concentrations are high in the CMOC case through changes in local circulation, due to invigoration of mixed-phase clouds from latent heat release. Although the results from this study are interesting and worthy of placement in the literature, there are a few issues that need to be resolved prior to publication in ACP.

**General comments:**

Although containing pertinent information, the introduction is somewhat difficult to follow. I suggest reordering and refocusing the introduction such that there are four paragraphs to guide the reader in a more efficient manner:

1. An abridged, broad background on aerosol-cloud-precipitation interactions, cloud phase, etc. Some of this information is already provided in the beginning of the introduction. Much of the information in the paragraph starting on p 5, l 73 could be placed in the first paragraph.

2. Introduce the concept behind CalWater and briefly describe previous relevant results, including the main findings from Ault et al. (2011), Creamean et al. (2013, 2014, 2015), White et al. (2015), Rosenfeld et al. (2013, 2014), and of course FAN2014.

3. Discuss what is missing from those previous works, as motivation for the current study. For instance, has anything been previously done regarding WMOC versus CMOS simulations? This seems to be a new approach that could be emphasized.

4. Clearly list the objectives for the current study and what is novel about it. The information on p 23, l 492-494 would be suitable for the list of objectives. Further, the authors state this is a follow up on FAN2014, but should specifically discuss what is new and why this is an improvement versus serving only as an extension (i.e., the information on p 9, l 168-172 and p 23, l 489-492 is an improvement that should be mentioned in the introduction).

*Creamean, J. M., Lee, C., Hill, T. C., Ault, A. P., DeMott, P. J., White, A. B., Ralph, F. M., and Prather, K. A.: Chemical properties of insoluble precipitation residue particles, J Aerosol Sci, 76, 13-27, 2014.*

*Creamean, J. M., Ault, A. P., White, A. B., Neiman, P. J., Ralph, F. M., Minnis, P., and Prather, K. A.: Impact of interannual variations in sources of insoluble aerosol species on orographic precipitation over California's central Sierra Nevada, Atmos Chem Phys, 15, 6535-6548, 2015.*

*Rosenfeld, D., Chemke, R., Prather, K., Suski, K., Comstock, J. M., Schmid, B., Tomlinson, J., and Jonsson, H.: Polluting of winter convective clouds upon transition from ocean inland over central California: Contrasting case studies, Atmos Res, 135, 112-127, 2014.*

*White, A. B., Neiman, P. J., Creamean, J. M., Coleman, T., Ralph, F. M., and Prather, K. A.: The Impacts of California's San Francisco Bay Area Gap on Precipitation Observed in the Sierra Nevada during HMT and CalWater, J Hydrometeorol, 16, 1048-1069, 2015.*

Even though the conditions for each case are described in FAN2014, they could be reiterated here. Some characteristics are presented on p 10, l 194-199, but what were the average cloud top and base heights? What was the frequency of occurrence for each cloud phase type and were the particular days chosen extremes? On p 10, l 192-193, I am assuming these averages for the case days only, but it would be interesting to provide information on if these are conditions that were anomalous or

typical of this region. Additionally, the description of the cases on p 23 l 494-497 would be better suited earlier on when describing the cases.

While a wide range of information is yielded from this more elaborate study, it is somewhat difficult to follow due to the nature in which the results are presented. As an example, the results quickly transition to comparing the CMOC to the WMOC case even before the basic results from the WMOC case are presented (p 17, l 356-366). I recommend reordering section 3 such that the CMOC results are presented first (section 3.1, without the subsections), WMOC second (section 3.2), followed by comparison of the microphysical changes from each case (i.e., section 3.1.3), and lastly a comparison on the disparate effects on precipitation from each case (i.e., section 3.1.2). Another option would be to condense and fold the comparison of the cases in terms of microphysical and precipitation effect differences in the discussion and conclusions. The authors could still focus on the CMOC case since it affords surprising results, but should be bolstered in the discussion. As a result, the figures would need to be restructured such that they are easier on the eye and align with the recommended reordering of section 3. For instance, Fig. 2 could instead be a combination of the current Fig. 2 and Fig. 3 panels, and Fig. 3 could be a combination of the current Fig. 4 and Fig. 5 panels for CMOC. The subsequent new figures (4 and 5) would then be the same structure, but for the WMOC case. The current Fig. 11 should be introduced with the WMOC case section (3.2). The current Figs. 6, 7, 8, 9, 10, and 12 would be pushed back to when the microphysical and precipitation accumulation differences are discussed. If restructured such that the results are reordered to enable better flow, the novelty of the work will be more apparent to the reader.

Publishing the new findings is key. To emphasize that this study entails new findings and is not a just a slight modification of FAN2014, the authors should consider providing specific statements as to how and why the results here vary from FAN2014 throughout the results section.

Along these lines, the fact that snow increases with increasing CCN is surprising. The authors present some comparison with previous work (i.e., Saleeby et al. (2011)) and what key differences may have led to the disparities between the studies. First, this should be done throughout the discussion: are the results (besides this one) surprising or expected in the context of previous work? Second, what other studies contradict this finding and why? The authors state that this result, "…is different from previous modeling studies in the literature…" but which studies specifically and for what reasons?

The authors do show the spatial heterogeneity in several resulting parameters in a couple figures, but are the main conclusions based upon the results time-dependent as well? For instance, CCN increasing snowfall, is that after (X) hours of simulation? Does this occur immediately? Or is this an average over the entire simulation time period, which could be highly variable over time? The authors could consider showing a figure of key parameters over time, which would be interesting.

It is not initially clear that the simulation parameters, namely CCN and INP concentrations, chosen are of realistic values to what is observed in the Sierra Nevada or if these are idealized situations. It is not until much later in the conclusions and discussion section that the authors mention CCN of > 1000 cm$^{-3}$ is considered an extreme for this region (p 26 l 554-555). This should be clearly delineated much earlier, in the methods. Also, what is "normal" versus extreme for the INP concentrations at the temperatures observed for each case?

There are several typos and grammatical mistakes throughout the manuscript, which the authors should take care in correcting for the revision. Some examples include: (1) "INP" is used in several instances where the plural form should be used (INPs), (2) CCN are plural but are commonly referred

to as a singular, and (3) "Mountains" is typically capitalized mid-sentence. Also, please write in past tense when describing the results from the simulations.

**Specific comments:**

Abstract: It is not apparent that the comparison of the WMOC and CMOC case are conducted under the same INP and CCN concentrations. Please clarify.

P 2, l 28: Please clarify the type of deposition (i.e., in-cloud nucleation, in-cloud scavenging, etc.).

P 2, l 30: "...WMOC *with* low INP *concentrations*." Also provide the INP concentration used here for reference.

P 2, l 30-31: Remove the sentence starting with "However" as this is redundant to the following sentence, which is better because it provides more detail. Once removed, the following sentence can be started with "*However,* we find a new mechanism..."

P 2, l 33: "...concentrations are > 1000 cm$^{-3}$."

P 2, l 34: Please clarify that this is the Central Valley and foothills west of the range.

P 2, l 33-37: There is quite a bit of information presented in this one sentence, making it appear as a run-on. The authors should consider breaking up into two sentences.

P 2, l 37: The beginning of this sentence is vague. What concentration of INPs? With what concentration of CCN? Some more context is needed.

P 2, l 39: "However, *an increase in precipitation occurs* in both cases..."

P 4, l 51: The Ralph et al. article on CalWater would be a great citation for this statement.

Ralph, F. M., Prather, K. A., Cayan, D., Spackman, J. R., DeMott, P., Dettinger, M., Fairall, C., Leung, R., Rosenfeld, D., Rutledge, S., Waliser, D., White, A. B., Cordeira, J., Martin, A., Helly, J., and Intrieri, J.: Calwater Field Studies Designed to Quantify the Roles of Atmospheric Rivers and Aerosols in Modulating Us West Coast Precipitation in a Changing Climate, B Am Meteorol Soc, 97, 1209-1228, 2016.

P 4, l 51-52: This sentence is redundant to that below, could simply remove.

P 4, l 54: Please clarify that this is over the Sierra Nevada mountains.

P 4, l 57: Cloud *phase* (should be singular). Please correct here and throughout.

P 4, l 65: Remove "in the atmosphere".

P 5, l 73: Be more specific by clarifying that these are aerosol *climate* impacts that depend on aerosol properties *such as number, size, and composition*.

Table 1 does not seem necessary. The information on the concentrations used are already provided in the text.

All figures: Why are two markers (circles) listed in the legend for INPs?

Fig. 2: Please place the panels in the order in which they are discussed in the text. Also, provide what the arrows are in the caption for clarity.

Fig. 6: Why are there no ice nucleation rates for levels where nucleated ice particles were found?

Figs. 8 and 9: Why is this only shown for CMOC and not WMOC? I get that the CMOC case presents interesting results, so at the very least, the authors could provide the WMOC spatial figures in a supporting document and allude to them in the text.

Fig. 9: It would be easier on the eye if a color scale much different than the previous figure were used, since these are differences and not absolute values. Perhaps red to white to blue?

---

## Referee Comment (RC2) · Anonymous Referee #2 · 13 Oct 2016

General Comments:

The response of cloud microphysical processes and precipitation to changes in aerosol particle concentration is still uncertain. This article presents numerical sensitivity tests on how the cloud processes and precipitation from mixed-phase orographic clouds are changed due to changes in the concentration of cloud condensation nuclei and ice forming nuclei. The results are interesting and are generally well presented. It should be publishable in ACP if the following specific issues could be considered in revision.

Specific Comments:

1) Line 51-53: Remove "Supercooled liquid occurred commonly in clouds over the Sierra Nevada during the cold season (Rosenfeld et al., 2013)", since the similar sentence also appears in line 54-55.

2) Line 67: "pollution aerosols" may be replaced by "anthropogenic aerosols".

3) Change Line 73-74 to "The impacts of aerosols on clouds not only depend on aerosols properties, but also on the dynamics and thermodynamics of the clouds".

4) Line 146: "which is referred to as INP concentration": this notation may not be proper, because the concentration of aerosol particles with diameter larger than 0.5 um is not the concentration of INP, just as a factor.

5) Line 166: The scheme for deposition nucleation should also be briefly described, since it dominates ice formation in the cold case.

6) Line 185: "...with the initial INP concentration of 0.1, 1, 10, and 100 cm-3, respectively": these are concentrations of coarse mode aerosol particles, not IN. This should be clarified.

7) Line 192: "... are around 30 (2) and 120 (4) cm-3, respectively": the concentrations of INPs should be the coarse mode aerosol particles. When we talk about the concentration of INP, we must indicate at which temperature.

8) Line 237-239: This is most likely caused by the treatment of snow particles in the model. Since most of the droplets transferred to snow when INP was high, the concentration and mass of water droplets must be lower. How the large drops are treated when they are frozen? Are they also transferred to snow?

9) Line 296: "...have ice nucleation occurring (Fig. 6b)": through which nucleation mechanism?

10) Line 372: "Atmospheric rivers" are mentioned several times, but it is not a commonly known concept. It should be explained at the beginning.

11) Line 401-404: It should not be the upper limit, if deposition nucleation and condensation freezing are not included.

12) Line 405-406: the CCN effect is much more significant than INP when the concentration of CCN is 1000 cm-3 or above.

13) Line 438-439: Remove "in our model simulation with the fast version of SBM in which ice habits are not considered".

14) Line 441-442: Remove "in the model simulation".

15) Page 42: The ordinates should be provided for Figure 10.

16) Page 43: The ordinates of the left panel should be provided for Figure 11.

17) Page 44: The unit of temperature in the figure should be corrected.

---

## Referee Comment (RC3) · A. Khain (Referee) · 21 Oct 2016

Review of the paper "Effects of cloud condensational nuclei and ice nucleating particles on precipitation processes and supercooled liquid in mixed-phase orographic clouds" , authored by J. Fan, L.R. Leung, D. Rosenfeld and P.J. DeMott.

The study presents a detailed analysis of process of ice formation and of precipitation response of orographic clouds over Sierra Nevada to the changes air temperature, CCN and IN. This study is an extension of the previous study by Fan et al. (2014). The strength of the study is the utilization WRF with spectral bin microphysics and wide use

budgets to evaluate rates and efficiency of one or another microphysical processes. The paper is of interest. I recommend to accept the paper with minor (from point of view of changes of the text), but important corrections. 1. Line 81. I suppose that reference to studies by: Lynn B., Khain, A. P., D. Rosenfeld, William L. Woodley, 2007: Effects of aerosols on precipitation from orographic clouds. Journal of Geophysical Research, 112, D10225 and to H. Noppel, A. Pokrovsky, B. Lynn , Khain, A. P., and K.D. Beheng 2010: On precipitation enhancement due to a spatial shift of precipitation caused by introducing small aerosols: numerical modeling. J. Geophys. Res.. 115, D18212, 17 PP., 2010, doi:10.1029/2009JD012645.

In both cases shift of precipitation by changing of CCN concentration was investigated.

2. Lines 152-158. Please describe the treatment of large AP clearer. Are these APs considered as CCN? Can these particles be activated to drops if S>0? What is soluble fraction of these large APs? (typically soluble fraction is about 0.1-0.2). Do you keep non-soluble fraction within the nucleated drops? 3. Line 160. Do you mean that you consider frozen drops as these large ice particles? Please add a more detailed explanation, even repeating some points from Fan et al. 2014. The paper should be self-consistent. 4. Line 166. What is the way of description of primary ice nucleation? Was it the same as in Khain et al. 2004, where the formula of Meyers et al was used? Or do you use formula by DeMott only for large APs that you consider as IN?

5. Line 182. Do you consider these large AP as IN separately from CCN? What is size of ice particles that form on the INP after its nucleation? What do you do with these AP if supersaturation over water is larger than zero? The questions 3-5 are caused by unclear description of IN treatment. 6. Line 548 and some places above. The statement is not correct. In the study by Lynn el al. (2007) mentioned above a dramatic increase in snow over mountains in case of high CCN concentration is reported and described in detail. In particular they presented figures 6-8 which are, in my opinion, similar to Fig 8 in the paper under revision. 7. Line 550. In the study by Lynn et al. 2007 it is shown that an increase in the AP concentration decreases warm rain

production and intensifies ice processes. The ice particles are advected downwind producing a substantial increase in snow and other ice precipitation over upwind slope and over the mountain peak. So the mechanism discussed in the study is not new and was described before. Besides, Lynn et al also discussed an important effect of very low relative humidity on the downwind slope. This low RH leads to evaporation of precipitating particles over downwind slope. As a result, effect of aerosols turned out to be also dependent on the wind speed because strong wind advected ice particles into zone of very low RH. So there is an "optimum" combination of APs concentration and wind speed to get maximum snow mass at the upwind slope and over the mountain peak. I propose that the authors discuss the similarities and differences of their results as compared with those reported by Lynn et al. (2007).

Please also note the supplement to this comment:
http://www.atmos-chem-phys-discuss.net/acp-2016-772/acp-2016-772-RC3-supplement.pdf

———————————————————

---

## Author Comment (AC1) · 6 Dec 2016

Fan et al. report on an aerosol-cloud-precipitation process modeling study regarding two cases from CalWater 2011. The advantage of this work over FAN2014 is based on the comparison of variable cloud phase conditions (WMOC versus CMOC), providing an added level of detail. One of the more surprising findings is the increase in snow precipitation when CCN concentrations are high in the CMOC case through changes in local circulation, due to invigoration of mixed-phase clouds from latent heat release. Although the results from this study are interesting and worthy of placement in the literature, there are a few issues that need to be resolved prior to publication in ACP.

Although containing pertinent information, the introduction is somewhat difficult to follow. I suggest reordering and refocusing the introduction such that there are four paragraphs to guide the reader in a more efficient manner: 1. An abridged, broad background on aerosol-cloud-precipitation interactions, cloud phase, etc.

- Thanks for the helpful suggestions to improve the paper. Please see our point-by-point responses as below.

**General comments:**

Some of this information is already provided in the beginning of the introduction. Much of the information in the paragraph starting on p 5, 73 could be placed in the first paragraph. 2. Introduce the concept behind CalWater and briefly describe previous relevant results, including the main findings from Ault et al. (2011), Creamean et al. (2013, 2014, 2015), White et al. (2015), Rosenfeld et al. (2013, 2014), and of course FAN2014. 3. Discuss what is missing from those previous works, as motivation for the current study. For instance, has anything been previously done regarding WMOC versus CMOS simulations? This seems to be a new approach that could be emphasized. 4. Clearly list the objectives for the current study and what is novel about it. The information on p23, 492-494 would be suitable for the list of objectives. Further, the authors state this is a follow up on FAN2014, but should specifically discuss what is new and why this is an improvement versus serving only as an extension (i.e., the information on p 9, 168-172 and p 23, 489-492 is an improvement that should be mentioned in the introduction).

*Creamean, J. M., Lee, C., Hill, T. C., Ault, A. P., DeMott, P. J., White, A. B., Ralph, F. M., and Prather, K. A.: Chemical properties of insoluble precipitation residue particles, J Aerosol Sci, 76, 13-27, 2014.*

*Creamean, J. M., Ault, A. P., White, A. B., Neiman, P. J., Ralph, F. M., Minnis, P., and Prather, K. A.: Impact of interannual variations in sources of insoluble aerosol species on orographic precipitation over California's central Sierra Nevada, Atmos Chem Phys, 15, 6535-6548, 2015.*

*Rosenfeld, D., Chemke, R., Prather, K., Suski, K., Comstock, J. M., Schmid, B., Tomlinson, J., and Jonsson, H.: Polluting of winter convective clouds upon transition*

*from ocean inland over central California: Contrasting case studies, Atmos Res, 135, 112-127, 2014.*

*White, A. B., Neiman, P. J., Creamean, J. M., Coleman, T., Ralph, F. M., and Prather, K. A.: The Impacts of California's San Francisco Bay Area Gap on Precipitation Observed in the Sierra Nevada during HMT and CalWater, J Hydrometeorol, 16, 1048-1069, 2015.*

- The Introduction generally follows the line that the reviewer suggested but starts with a general background about California precipitation and cloud properties that this study focuses on. Then the factors – AR and aerosols that impact cloud properties and precipitation are introduced in the second paragraph. The third and fourth paragraphs basically follow the second one to give a more detailed literature survey about aerosol impact on orographic clouds and supercooled water.  After that, we discuss what is missing from those previous studies, and introduce FAN2014, and state the objectives of this study following FAN2014. All authors are in agreement that the Introduction of this paper is organized in a logical fashion to introduce the topic and goals of this study.
– We have included the references that the reviewer suggested except White et al. (2015), which we think is not much related (see L75-77 and L90). We have also slightly modified the text to more specifically discuss what is missing from the previous studies and what is new in this study (i.e., the text at L100-102, L122-123, and L129-130).
- The text on p23, L492-494 in the original manuscript was already stated in our objectives #2 and #3 (i.e., the current L147-150), and the information on p 9, L168-172 was also already included in the introduction (i.e., the last sentence of the Section 1).

Even though the conditions for each case are described in FAN2014, they could be reiterated here. Some characteristics are presented on p 10, 194-199, but what were the average cloud top and base heights? What was the frequency of occurrence for each cloud phase type and were the particular days chosen extremes? On p 10, 192-193, I am assuming these averages for the case days only, but it would be interesting to provide information on if these are conditions that were anomalous or typical of this region. Additionally, the description of the cases on p 23 494-497 would be better suited earlier on when describing the cases.

- The averaged cloud top height for each cloud case is described on P5-6 when the cases are introduced for the first time. Now we have reiterated here (p10 L209-211). The cloud base information has been added as well (L212-213). Those two cases correspond to anomalous conditions as they are influenced by both AR and long-range transported dust/bio (L214). The description of the cases later on p 23 494-497 (the original version) is just a short version of the description here but with a little different wordings. Now we have added the same wordings (L217-218).

While a wide range of information is yielded from this more elaborate study, it is somewhat difficult to follow due to the nature in which the results are presented. As an example, the results quickly transition to comparing the CMOC to the WMOC case even before the basic results from the WMOC case are presented (p 17, l 356-366). I recommend reordering section 3 such that the CMOC results are presented first (section 3.1, without the subsections), WMOC second (section 3.2), followed by comparison of

the microphysical changes from each case (i.e., section 3.1.3), and lastly a comparison on the disparate effects on precipitation from each case (i.e., section 3.1.2). Another option would be to condense and fold the comparison of the cases in terms of microphysical and precipitation effect differences in the discussion and conclusions. The authors could still focus on the CMOC case since it affords surprising results, but should be bolstered in the discussion. As a result, the figures would need to be restructured such that they are easier on the eye and align with the recommended reordering of section 3. For instance, Fig. 2 could instead be a combination of the current Fig. 2 and Fig. 3 panels, and Fig. 3 could be a combination of the current Fig. 4 and Fig. 5 panels for CMOC. The subsequent new figures (4 and 5) would then be the same structure, but for the WMOC case. The current Fig. 11 should be introduced with the WMOC case section (3.2). The current Figs. 6, 7, 8, 9, 10, and 12 would be pushed back to when the microphysical and precipitation accumulation differences are discussed. If restructured such that the results are reordered to enable better flow, the novelty of the work will be more apparent to the reader.

- Section 3 is already organized in the way that the reviewer suggests: CMOC is discussed first (Section 3.1) and then WMOC (Section 3.2). Within each case, we discuss the basic results first and then look into the mechanisms.  We do not quite understand the reviewer's comment "the results quickly transition to comparing the CMOC to the WMOC case even before the basic results from the WMOC case are presented". The place that the reviewer pointed out (p 17, L356-366 in the previous version) only contains one sentence that mentions a sensitivity test done based on WMOC to confirm a factor in the mechanism we presented, but that is after all the basic results and the mechanisms for the CMOC have been presented and discussed. We discussed that sensitivity test here in order to give a whole picture of the mechanism at the same place. The WMOC is chosen for this test because of less mixed-phase regime compared with CMOC, so the factor would have a more significant role in the CMOC if it plays a role in the WMOC. This has been clarified further on P17 L377-381 in the current manuscript. The subsection titles are useful for the readers to follow the result section clearly.
- The sequence of the figures is also already presently constructed around the logical discussion of research findings: starting from the significant results/concerns in precipitation (Fig 2), then looking into how they are related to cloud microphysical properties (Fig 3), which are determined by major microphysical process rates (i.e., budgets; Fig 4, and 5.). After that, then we present the physical mechanisms leading to the significant changes (Fig. 6-Fig. 10).  So, they are separately presented in their natural ways, e.g., Fig.2 is about precipitation and Fig. 3 is about cloud microphysical properties. Besides, each of the figures has multiple panels already. We prefer not to combine the figures as the reviewer suggested.
- Moving Figures 6, 7, 8, 9, 10, and 12 to the end does not align with the flow of the presentation in the paper, as these follow the discussion logically. In addition, it is quite common to introduce figures with comparative results prior to completing their discussion (i.e., they are referred back to), and this goes along naturally with the manner of discussing CMOC first and WMOC second.

Publishing the new findings is key. To emphasize that this study entails new findings and is not a just a slight modification of FAN2014, the authors should consider providing

specific statements as to how and why the results here vary from FAN2014 throughout the results section.

- All the results discussed in the result section are new from FAN2014. We do not think that it is necessary to say it throughout the results section. The only thing that we can compare is that the significant CCN impacts on precipitation was not seen in FAN2014, which was simply because we only increased CCN by 3 times based on the baseline cases in FAN2014, making CCN concentrations of ~ 160 and 720 cm$^{-3}$ in the high CCN cases for CMOC and WMOC, respectively. They are smaller than 1000 cm$^{-3}$ where the significant effect is seen in this study. In addition, CCN and IN are set to be uniform and increased uniformly over the domain, while in FAN2014, only CCN over the central valley and coastal urban area were increased. This discussion has been added to the last section (P25,L551-556).

Along these lines, the fact that snow increases with increasing CCN is surprising. The authors present some comparison with previous work (i.e., Saleeby et al. (2011)) and what key differences may have led to the disparities between the studies. First, this should be done throughout the discussion: are the results (besides this one) surprising or expected in the context of previous work? Second, what other studies contradict this finding and why? The authors state that this result, "...is different from previous modeling studies in the literature..." but which studies specifically and for what reasons?

- Such comparisons with literature studies were already presented in the Discussion section in two places (starting from L572 on P25 and the last paragraph of the paper). For the sentences that the reviewer pointed out, we have provided the possible reasons and modified that text as "different from previous modeling studies in the literature such as Lowenthal et al. (2011). Many possible reasons could lead to the differences including different cloud cases and different model parameterizations especially for riming processes" (P26 L575-577). In addition, we have provided more detailed discussion by comparing with other previous studies brought up by another reviewer as shown in L583-596.

The authors do show the spatial heterogeneity in several resulting parameters in a couple figures, but are the main conclusions based upon the results time-dependent as well? For instance, CCN increasing snowfall, is that after (X) hours of simulation? Does this occur immediately? Or is this an average over the entire simulation time period, which could be highly variable over time? The authors could consider showing a figure of key parameters over time, which would be interesting.

- Yes, time evolution is important to look at the mechanism responsible for the changes. But we already considered this information and discussed when the precipitation (or snow) enhancement starts, and then looked at the related variables at the start time and how they evolve in the subsequent 1-2 hours as shown in Fig. 6-10. The corresponding text to discuss this starts from "Since the precipitation enhancement begins at 1400 UTC, which is a couple of hours into the simulations, we focus on the time period of 14-1600 UTC" (L342-344 in the current manuscript).

It is not initially clear that the simulation parameters, namely CCN and INP concentrations, chosen are of realistic values to what is observed in the Sierra Nevada or if these are idealized situations. It is not until much later in the conclusions and discussion section that the authors mention CCN of > 1000 cm$^{-3}$ is considered an extreme for this region (p 26 l 554-555). This should be clearly delineated much earlier, in the methods. Also, what is "normal" versus extreme for the INP concentrations at the temperatures observed for each case?

- We agree that it is useful to further frame the INP concentration range used and have added Table 2 and text to the present discussion of how dust/bio particle concentrations relate to INP concentrations as a function of temperature. This discussion has been added on P9 L196-205. The extreme conditions for CCN and INP were mentioned when the simulation setup was introduced (now L190-191).

There are several typos and grammatical mistakes throughout the manuscript, which the authors should take care in correcting for the revision. Some examples include: (1) "INP" is used in several instances where the plural form should be used (INPs), (2) CCN are plural but are commonly referred to as a singular, and (3) "Mountains" is typically capitalized mid-sentence. Also, please write in past tense when describing the results from the simulations.

- We have carefully checked these places and corrected the typos and grammatical mistakes throughout the paper.

Abstract: It is not apparent that the comparison of the WMOC and CMOC case are conducted under the same INP and CCN concentrations. Please clarify.

- We have modified a sentence in Abstract to clearly say it, i.e., to "We quantify the CCN and INP impacts on supercooled water content, cloud phases and precipitation for a WMOC and a CMOC case with sensitivity tests using the same CCN and INP concentrations between the WMOC and CMOC"

**Specific comments:**

P 2, l 28: Please clarify the type of deposition (i.e., in-cloud nucleation, in-cloud scavenging, etc.).

- By deposition we are referring to the depositional ice growth process.

P 2, l 30: "...WMOC *with* low INP *concentrations*." Also provide the INP concentration used here for reference.

-As we have clarified in the current version, INPs are dependent of temperature besides dust/bio concentrations. So, it is not just one value that can be put there. .

P 2, l 30-31: Remove the sentence starting with "However" as this is redundant to the following sentence, which is better because it provides more detail. Once removed, the following sentence can be started with "*However,* we find a new mechanism..."

- Since the new mechanism is to explain the sentence "this reverses strongly for CCN > 1000 cm$^{-3}$", we think putting the word "however" before this sentence better conveys what we want to say here.

P 2, l 33: "...concentrations are > 1000 cm$^{-3}$."

- We have revised it as "for CCN of 1000 cm$^{-3}$ and larger".

P 2, l 34: Please clarify that this is the Central Valley and foothills west of the range.

- Done.

P 2, l 33-37: There is quite a bit of information presented in this one sentence, making it appear as a run-on. The authors should consider breaking up into two sentences.

- We have re-constructed the sentences by breaking into short pieces. And now it reads as "In this situation, more widespread shallow clouds with greater amount of cloud water form in the Central Valley and foothills west of the mountain range. The increased latent heat release associated with the formation of these clouds strengthens the local transport of moisture to the windward slope, invigorating mixed-phase clouds over the mountains, and thereby producing higher amounts of snow precipitation." (P2 L34-38).

P 2, l 37: The beginning of this sentence is vague. What concentration of INPs? With what concentration of CCN? Some more context is needed.

- This is a generalized summary of details that can only be understood through reading the paper. We have added "under all CCN conditions" to the sentence.

P 2, l 39: "However, *an increase in precipitation occurs* in both cases..."

- Changed.

P 4, l 51: The Ralph et al. article on CalWater would be a great citation for this statement.

Ralph, F. M., Prather, K. A., Cayan, D., Spackman, J. R., DeMott, P., Dettinger, M., Fairall, C., Leung, R., Rosenfeld, D., Rutledge, S., Waliser, D., White, A. B., Cordeira, J., Martin, A., Helly, J., and Intrieri, J.: Calwater Field Studies Designed to Quantify the Roles of Atmospheric Rivers and Aerosols in Modulating Us West Coast Precipitation in a Changing Climate, B Am Meteorol Soc, 97, 1209-1228, 2016.

- Added.

P 4, l 51-52: This sentence is redundant to that below, could simply remove.

- Removed.

P 4, l 54: Please clarify that this is over the Sierra Nevada mountains.

- Done.

P 4, l 57: Cloud *phase* (should be singular). Please correct here and throughout.

- Since cloud has different phases (liquid, mixed, and ice phases) and it has been used as plural commonly as well.

P 4, l 65: Remove "in the atmosphere".

- Done.

P 5, l 73: Be more specific by clarifying that these are aerosol *climate* impacts that depend on aerosol properties *such as number, size, and composition*.

- Added "such as number, size, and composition".

Table 1 does not seem necessary. The information on the concentrations used are already provided in the text.

- Table 1 more clearly and intuitively shows what kind of simulations we have conducted for this study than text.

All figures: Why are two markers (circles) listed in the legend for INPs?

- We listed two markers to show the legend more clearly.

Fig. 2: Please place the panels in the order in which they are discussed in the text. Also, provide what the arrows are in the caption for clarity.

- Fig. 2b and 2c follow the conventional presentation on warm and ice precipitation respectively. To be consistent with Fig.14 that is the same type of figure, we would like to keep it this way. This is allowed in scientific papers (i.e., Fig. 2c is referred before Fig. 2b). We have added the description of the two arrows to the Figure caption now.

Fig. 6: Why are there no ice nucleation rates for levels where nucleated ice particles were found?

- Fig. 6c is in logarithm scale and the nucleation rate is too low below 2.5 km altitude, so it is not shown.

Figs. 8 and 9: Why is this only shown for CMOC and not WMOC? I get that the CMOC case presents interesting results, so at the very least, the authors could provide the WMOC spatial figures in a supporting document and allude to them in the text.

- Fig. 8-9 are parts of the figures illustrating the mechanism leading to the drastic CCN impacts on precipitation in the CMOC. Since very similar mechanisms are seen in the WMOC, there is no need to present similar figures but in the text we clearly discuss the similarity starting from L490 on P20, "We have done the same investigation as in Section 3.1.1, and found the mechanism causing the increased cloud water and the snow

production is similar as that in CMOC, that is, …". Furthermore, the key part of the mechanism for the WMOC is shown in Fig. 10b, which is the change of local circulation that increases the zonal transport of moisture to the windward slope of the mountains.

Fig. 9: It would be easier on the eye if a color scale much different than the previous figure were used, since these are differences and not absolute values. Perhaps red to white to blue?

- Right now it is red to green to blue, not much different from what the reviewer suggested (i.e., red to white to blue), and we feel that it shows the positive and negative data clearly.

---

## Author Comment (AC2) · 6 Dec 2016

Response to Reviewer #2

General Comments:

The response of cloud microphysical processes and precipitation to changes in aerosol particle concentration is still uncertain. This article presents numerical sensitivity tests on how the cloud processes and precipitation from mixed-phase orographic clouds are changed due to changes in the concentration of cloud condensation nuclei and ice forming nuclei. The results are interesting and are generally well presented. It should be publishable in ACP if the following specific issues could be considered in revision.

- Thanks for the helpful suggestions to improve the paper. Please see our point-by-point responses as below.

Specific Comments:

1) Line 51-53: Remove "Supercooled liquid occurred commonly in clouds over the Sierra Nevada during the cold season (Rosenfeld et al., 2013)", since the similar sen- tence also appears in line 54-55.

-Done.

2) Line 67: "pollution aerosols" may be replaced by "anthropogenic aerosols".

-Done.

3) Change Line 73-74 to "The impacts of aerosols on clouds not only depend on aerosols properties, but also on the dynamics and thermodynamics of the clouds".

- Changed to "Aerosol impacts on clouds not only depend on aerosol properties, but also dynamics and thermodynamics".

4) Line 146: "which is referred to as INP concentration": this notation may not be proper, because the concentration of aerosol particles with diameter larger than 0.5 um is not the concentration of INP, just as a factor.

- We apologize for the confusion in using the term of INP. Besides deleting "which is referred to as INP concentration" here, we have clarified at L156-158, "…so in this paper we vary the constant $n_{a>0.5\mu m}$ over a range of relevant conditions to investigate the impacts of varied INP concentration".

5) Line 166: The scheme for deposition nucleation should also be briefly described, since it dominates ice formation in the cold case.

- FAN2014 detailed why deposition/condensation freezing is not included. Adding deposition/condensation freezing produces large amount of small ice particles that are not observed for those cases. We have added the sentence "Adding deposition/condensation freezing produces large amount of small ice particles, which is not consistent with

observations, thereby deposition/condensation freezing is not included, as discussed in FAN2014" (P8 L164-166).

6) Line 185: ". . .with the initial INP concentration of 0.1, 1, 10, and 100 cm-3, respectively": these are concentrations of coarse mode aerosol particles, not IN. This should be clarified.

- Yes, we have changed to dust/bio (or INP proxy) throughout the paper. We have also further frame the INP concentration range used and have added to the present discussion of how dust/bio particle concentrations relate to INP concentrations as a function of temperature (P9 L196-203).

- In addition, in our model, INP proxy concentration is a single prognostic variable separately from CCN. For the simulation of the observed case in FAN2014, it is initiated with the concentrations of clear-sky aerosol particles with diameter larger than 0.5 μm in the dust layer. Since our model does not have full aerosol simulations and INP proxy concentration in our model is not a factor from the predicted aerosol simulations, saying "they are the concentrations of coarse mode aerosol particles" would confuse people. We have added text to clarify, i.e., "As described in FAN2014, dust/bio particle concentration (i.e., IN proxies) is a single prognostic variable separate from CCN. For the simulation of the observed case in FAN2014, dust/bio concentration is initiated with the concentration of clear-sky aerosol particles with diameter larger than 0.5 μm in the dust layer" (L172-175).

7) Line 192: ". . . are around 30 (2) and 120 (4) cm-3, respectively": the concentra- tions of INPs should be the coarse mode aerosol particles. When we talk about the concentration of INP, we must indicate at which temperature.

- This comment is the same as #6. Please see our response above.

8) Line 237-239: This is most likely caused by the treatment of snow particles in the model. Since most of the droplets transferred to snow when INP was high, the concentration and mass of water droplets must be lower. How the large drops are treated when they are frozen? Are they also transferred to snow?

- It could be. But even the ice nucleation forms mainly cloud ice, the large amount of ice could lead to lower rain concentrations due to conversion through WBF and riming. Yes, large droplets are transferred to snow when immersion freezing occurs (cloud ice and snow are represented with one set of size bins and distinguished with a radius of 150 microns).

9) Line 296: "...have ice nucleation occurring (Fig. 6b)": through which nucleation mechanism?

- Only immersion freezing from DeMott et al. 2015 is considered in the simulations, as stated in the model setup and mentioned in the paragraph above. We have also mentioned specifically here (L348 in the current manuscript).

10) Line 372: "Atmospheric rivers" are mentioned several times, but it is not a commonly known concept. It should be explained at the beginning.

- In fact, the terminology is commonly known and is the dynamical and thermodynamic environment for precipitation events over the western US. This is explained at the beginning (L72-74).

11) Line 401-404: It should not be the upper limit, if deposition nucleation and condensation freezing are not included.

- As we discussed in FAN2014 and earlier in this paper, deposition/condensation nucleation should not be the case for those clouds since it forms a great amount of small ice crystals that were not observed by aircraft measurements. In addition, deposition/condensation nucleation does not directly convert liquid to ice, and it competes for INPs with immersion freezing (less immersion freezing will occur if deposition/condensation nucleation occurs). So, the largest effect on the SCW and cloud phase should be through immersion freezing with a given INP condition.

12) Line 405-406: the CCN effect is much more significant than INP when the concentration of CCN is 1000 cm-3 or above.

- That was for precipitation. For the liquid fraction discussed here, we do not see that as shown in Fig. 12, although the more significant CCN effect is seen for CCN > 300 cm$^{-3}$, but it is still smaller compared with the INP effect.

13) Line 438-439: Remove "in our model simulation with the fast version of SBM in which ice habits are not considered".

- In fact this was added to address one of the comments from a coauthor because the HM processes are sensitive to ice habits in nature but this fast version of SBM does not consider this in the HM processes.

14) Line 441-442: Remove "in the model simulation".

- We prefer to keep it since recent observations suggested secondary ice nucleation could be significant but the model might not able to simulate it. There might be additional secondary ice nucleation mechanisms besides HM processes, or the parameterization of HM processes is not adequate.

15) Page 42: The ordinates should be provided for Figure 10.

- The ordinates have been refined.

16) Page 43: The ordinates of the left panel should be provided for Figure 11.

- The ordinates have been refined.

17) Page 44: The unit of temperature in the figure should be corrected.

- Sorry we forgot to mark that the grey contour lines are the geophysical height in meters. They are not temperatures. It has been noted in the current figure caption.

---

## Author Comment (AC3) · 6 Dec 2016

Review of the paper "Effects of cloud condensational nuclei and ice nucleating particles on precipitation processes and supercooled liquid in mixed-phase orographic clouds" , authored by J. Fan, L.R. Leung, D. Rosenfeld and P.J. DeMott.

The study presents a detailed analysis of process of ice formation and of precipitation response of orographic clouds over Sierra Nevada to the changes air temperature, CCN and IN. This study is an extension of the previous study by Fan et al. (2014). The strength of the study is the utilization WRF with spectral bin microphysics and wide use budgets to evaluate rates and efficiency of one or another microphysical processes. The paper is of interest. I recommend to accept the paper with minor (from point of view of changes of the text), but important corrections.

- Prof. Khain, thank you so much for the useful comments to improve our manuscript. Please see our detailed responses below.

1. Line 81. I suppose that reference to studies by: Lynn B., Khain, A. P., D. Rosenfeld, William L. Woodley, 2007: Effects of aerosols on precipitation from orographic clouds. Journal of Geophysical Research, 112, D10225 and to H. Noppel, A. Pokrovsky, B. Lynn , Khain, A. P., and K.D. Beheng 2010: On precipitation enhancement due to a spatial shift of precipitation caused by introducing small aerosols: numerical modeling. J. Geophys. Res.. 115, D18212, 17 PP., 2010, doi:10.1029/2009JD012645.

In both /cases shift of precipitation by changing of CCN concentration was investigated.

- Sorry that we forgot to cite Lynn et al. (2007) at this place since it was the first study showing the spillover effects. We have added it now. Noppel et al. (2010) does not fit here since we are discussing aerosol impacts on orographic precipitation here. But we have discussed this paper in the last section at L626-631 (P. 25-26).

2. Lines 152-158. Please describe the treatment of large AP clearer. Are these APs considered as CCN? Can these particles be activated to drops if S>0? What is soluble fraction of these large APs? (typically soluble fraction is about 0.1-0.2). Do you keep non-soluble fraction within the nucleated drops?

- We think you might have misunderstood the statements here. These descriptions are about freezing of large liquid drops through immersion freezing, not about freezing of aerosol particles.

3. Line 160. Do you mean that you consider frozen drops as these large ice particles? Please add a more detailed explanation, even repeating some points from Fan et al. 2014. The paper should be self-consistent.

- The size of formed ice particles through immersion freezing depends on drop size. Our implementation of the immersion freezing starts from the largest drops freeze first, followed by the smaller ones over the size spectrum of water drops when immersion freezing occurs. Therefore, this implementation yields relatively large ice particles in the model simulation, which is consistent with observations. Deposition/condensation ice nucleation is not considered in this study since it produced large amount of small ice particles that were not observed. We have changed the text here and also repeated some text in FAN2014. Now it is read as "An added feature of the implementation is that when immersion freezing occurs, freezing starts from the largest drops first, followed by the smaller ones over the size spectrum of water drops. This implementation yielded the majority of large ice particles as observed by aircraft measurements (FAN2014). Adding deposition/condensation freezing produces a large amount of small ice particles, which is not consistent with observations, so deposition/condensation freezing is not included, as discussed in FAN2014. The assumption that the largest drops freeze first also acknowledges the expectation that the largest droplets should have a higher probability of containing an INP active at a given temperature" (L161-168)

4. Line 166. What is the way of description of primary ice nucleation? Was it the same as in Khain et al. 2004, where the formula of Meyers et al was used? Or do you use formula by DeMott only for large APs that you consider as IN?

- We did not consider deposition/condensation freezing, so Meyers et al 1992 is not included. The reason was discussed in FAN2014 and now has been added here in L164-166. INP is a single prognostic variable separately from CCN. For the simulation of the observed case in FAN2014, dust/bio is initiated with the concentrations of clear-sky aerosol particles with diameter larger than 0.5 μm in the dust layer. The relevant text is at L150-154 and L172-176. For examining the impacts of INP, we change the initial dust/bio particle concentration of 0.1, 1, 10, and 100 cm$^{-3}$, respectively, referred to as IN0.1, IN1, IN10, and IN100 (L196-198). We have also added Table 2 and some text to the present discussion of how dust/bio particle concentrations relate to INP concentrations as a function of temperature based on DeMott et al. (2015) on P9 L196-205.

5. Line 182. Do you consider these large AP as IN separately from CCN? What is size of ice particles that form on the INP after its nucleation? What do you do with these AP if supersaturation over water is larger than zero? The questions 3-5 are caused by unclear description of IN treatment.

- Our responses to Comments 3-4 should have addressed the questions here.

6. Line 548 and some places above. The statement is not correct. In the study by Lynn el al. (2007) mentioned above a dramatic increase in snow over mountains in case of high CCN concentration is reported and described in detail. In particular they presented

figures 6-8 which are, in my opinion, similar to Fig 8 in the paper under revision.

- Here we are talking about the mechanism leading to the drastically increased snow precipitation on the windward slope of the mountain, which is new indeed. For your convenience, we repeat the mechanism here as below,

Increasing CCN forms more shallow clouds at the wide valley area and foothills, which induces a change of local circulation through more latent heat release and increases the zonal transport of moisture to the windward slope of the mountains. This results in much more invigorated mixed-phase clouds with enhanced deposition and riming processes and therefore much more snow precipitation.

Lynn et al. (2007) indeed showed the increased snow water content in cloud by CCN but not snow precipitation on the windward slope of the mountain. That study showed decreased precipitation on the windward slope in the polluted case and increased precipitation over the downwind slope (with a decreased total precipitation). Their explanation for the increased snow is through collision of ice particles formed from the enhanced drop freezing due to delayed warm rain formation. So, as you can see, the results and the mechanism are different from our study. We have added the relevant discussion of Lynn et al. 2007 here, that is, "Lynn et al. (2007) also showed that increasing small aerosol particles led to an increased in-cloud snow mass content as a result of more ice particles formed from droplet freezing due to suppressed warm rain formation and thereby more collisions between those ice particles. But different from our study, the total precipitation on the windward slope in Lynn et al. (2007) was decreased as the snow particles had smaller size with lower fall speeds, and they were advected to the lee-side of the mountain, resulting in more precipitation there" (L585-590).

7. Line 550. In the study by Lynn et al. 2007 it is shown that an increase in the AP concentration decreases warm rain production and intensifies ice processes. The ice particles are advected downwind producing a substantial increase in snow and other ice precipitation over upwind slope and over the mountain peak. So the mechanism discussed in the study is not new and was described before. Besides, Lynn et al also discussed an important effect of very low relative humidity on the downwind slope. This low RH leads to evaporation of precipitating particles over downwind slope. As a result, effect of aerosols turned out to be also dependent on the wind speed because strong wind advected ice particles into zone of very low RH. So there is an "optimum" combination of APs concentration and wind speed to get maximum snow mass at the upwind slope and over the mountain peak. I propose that the authors discuss the similarities and differences of their results as compared with those reported by Lynn et al. (2007).

- Lynn et al. (2017) showed that increased CCN decreased precipitation on the upwind slope. The increased precipitation occurred over the downwind slope (not the upwind slope), which is very different from our result of drastically increased snow precipitation on the upwind slope through a mechanism of changed circulation that enhances the transport of moisture from the valley to the mountain.  As discussed in our response to the comment right above, they are very different results and mechanisms. We have discussed the similarities and differences of the results between Lynn et al. (2007) and

our study and the possible reasons for the differences, as shown on P26 from L585 to L598.

- We agree that the aerosol impacts would depend on dynamics (wind speed) and thermodynamics (RH), as studied in Lynn et al. (2007). Although we did not carry out such sensitivity tests, we showed similar results and mechanisms in two cases with very different wind direction and RH. We have added discussion about this as shown in L599-603, "The mechanism leading to the enhanced precipitation over the windward slope by increasing CCN is seen in the two cases with very different cloud temperature, wind direction and RH. However, the efficiency of the mechanism could depend on dynamics (wind speed) and thermodynamics (RH). As examined in Lynn et al. (2007), aerosol impact on the orographic precipitation is reduced when RH is very high and increased as wind speed is reduced".